# Rapid expansion of Treg cells protects from collateral colitis following a viral trigger

Michelle Schorer[1], Katharina Lambert[1,7], Nikolas Rakebrandt[1], Felix Rost[1], Kung-Chi Kao[1], Alexander Yermanos[2,3], Roman Spörri [2], Josua Oderbolz[2], Miro E. Raeber [4], Christian W. Keller [1,8], Jan D. Lünemann [1,8], Gerhard Rogler[5], Onur Boyman [4,6], Annette Oxenius [2] & Nicole Joller [1✉]

Foxp3[+] regulatory T (Treg) cells are essential for maintaining peripheral tolerance and preventing autoimmunity. While genetic factors may predispose for autoimmunity, additional environmental triggers, such as viral infections, are usually required to initiate the onset of disease. Here, we show that viral infection with LCMV results in type I IFN-dependent Treg cell loss that is rapidly compensated by the conversion and expansion of Vβ5[+] conventional T cells into iTreg cells. Using Vβ5-deficient mice, we show that these Vβ5[+] iTreg cells are dispensable for limiting anti-viral immunity. Rather, the delayed replenishment of Treg cells in Vβ5-deficient mice compromises suppression of microbiota-dependent activation of CD8[+] T cells, resulting in colitis. Importantly, recovery from clinical symptoms in IBD patients is marked by expansion of the corresponding Vβ2[+] Treg population in humans. Collectively, we provide a link between a viral trigger and an impaired Treg cell compartment resulting in the initiation of immune pathology.

[1] Institute of Experimental Immunology, University of Zurich, Winterthurerstrasse 190, 8057 Zurich, Switzerland. [2] Institute of Microbiology, ETH Zurich, Vladimir-Prelog-Weg 1-5/10, 8093 Zurich, Switzerland. [3] Laboratory for Systems and Synthetic Immunology, D-BSSE, ETH Zurich, 4058 Basel, Switzerland. [4] Department of Immunology, University Hospital Zurich, Häldeliweg 4, 8044 Zurich, Switzerland. [5] Department of Gastroenterology and Hepatology, University Hospital Zurich, Rämistrasse 100, 8091 Zurich, Switzerland. [6] Faculty of Medicine, University of Zurich, Pestalozzistrasse 3/5, 8091 Zurich, Switzerland. [7] Present address: Benaroya Research Institute at Virginia Mason, Seattle, WA 98101, USA. [8] Present address: Department of Neurology with Institute of Translational Neurology, University Hospital Munster, 48149 Munster, Germany. ✉email: nicole.joller@immunology.uzh.ch

Regulatory T cells (Treg cells) expressing the transcription factor FOXP3 represent a unique population of CD4[+] T helper cells that are essential for maintaining immune homeostasis. Treg cells ensure peripheral tolerance by controlling autoreactive T cells that have escaped thymic selection and their loss results in overt autoimmune disease as seen in patients with immunodysregulation polyendocrinopathy enteropathy X-linked (IPEX) syndrome, which lack functional Treg cells due to a mutation in *FOXP3*[1]. Moreover, lack of Treg-mediated immune control is a key feature of numerous autoimmune diseases and their models[2,3]. Specifically, several autoimmune conditions, including rheumatoid arthritis (RA), systemic lupus erythematosus (SLE), and inflammatory bowel disease (IBD), are characterized by a relative loss of Treg cells or their functional impairment[4–9]. However, what triggers this loss or functional impairment is still unclear.

Viral infections have been suggested as triggers of autoimmunity and a number of autoimmune diseases show a strong linkage to certain viral infections[8,10,11]. Furthermore, viral gastrointestinal infections have been associated with a higher likelihood to develop IBD later in life[12]. Viral infection triggers robust anti-viral immunity that is often dominated by a strong interferon (IFN) response to mediate viral control. However, particularly type I IFNs might play an ambivalent role in this context. On the one hand they represent a powerful immune defense mechanism against the viral infection[13,14], on the other hand they play an important role in exacerbating autoimmune diseases, such as SLE or RA[15–17]. Interestingly, long-term IFNα therapy as used in cancer and hepatitis patients can precipitate SLE and systemic sclerosis[18–20]. Type I IFNs support a vast array of protective host defense mechanisms including the restriction of viral replication[14], NK cell cytotoxicity[21], and survival of cytotoxic T cells[22]. In addition, recent studies revealed that type I IFN receptor (IFNAR) engagement also impacts on Treg cell activation and proliferation under inflammatory conditions[23,24]. However, how viral infections and the concomitant type I IFN response might affect the ability of Treg cells to maintain peripheral tolerance has not been addressed.

Using lymphocytic choriomeningitis virus (LCMV) infection as a model, we investigated how the Treg cell compartment is affected during viral infections that induce a strong type I IFN response. Our results show that the transient loss of Treg cells, mediated by type I IFNs, is rapidly compensated by induced Treg (iTreg) cells. This Treg population is dominated by a single β chain TCR, Vβ5. Infection of Vβ5-deficient mice revealed that absence of Vβ5[+] Treg cells has no direct effect on anti-viral immunity. In contrast, Vβ5[+] Treg cells are essential for restraining an underlying colitogenic CD8[+] T cell response. This colitis is triggered by microbial antigens released due to increased intestinal permeability resulting from the viral infection. We found that Vβ5[+] T cells harbour a higher potential to convert into Tregs and display enhanced proliferation as they receive an intrinsically elevated TCR signal. This also manifests in an overrepresentation of Vβ5[+] T cells within the Treg pool at steady-state. We also identify Vβ2[+] T cells as the corresponding population in humans that is overrepresented in colitis patients with inactive disaese. In summary, we demonstrate that the virus-induced type I IFN response transiently compromises the Treg compartment and that a rapid compensation through conversion of Vβ5[+] T cells into Treg cells is necessary to prevent autoimmune inflammation.

## Results

### iTregs compensate transient Treg loss upon viral infection.

To determine how viral infections and the ensuing immune response affect the Treg compartment, we used LCMV as a model for a viral infection eliciting a potent type I IFN response[25]. We used LCMV strain WE to induce acute and LCMV strain clone 13 to induce chronic infection[26,27]. In line with previous studies[23,24], we observed a pronounced loss in Treg cells at the peak of immune response upon LCMV infection (day 5–7; Fig. 1a), coinciding with marked weight loss (Supplementary Fig. 1a)[28,29]. However, this loss in Treg cells was only transient as the the Treg niche was readily replenished upon recovery, predominantly with CXCR3[+]Nrp1[−] iTreg cells (Fig. 1b, Supplementary Fig. 1b, c). While both acute and chronic LCMV infection resulted in an expansion of CXCR3[+] iTreg cells, it was most pronounced upon chronic LCMV infection, where peripherally induced CXCR3[+]Nrp1[−] iTreg cells accounted for over half of the Treg population (Fig. 1c).

Depending on the strain, LCMV can cause acute or chronic infections with varying intensities of anti-viral type I IFN responses[26,27]. To assess the Treg composition in these different scenarios, we infected mice with a range of LCMV strains that result in acute (Armstrong, WE) or chronic (Clone 13, Docile) infection. Analyzing the TCR composition by flow cytometry, we noticed that all LCMV strains selectively enriched for Vβ5.1,5.2[+] Treg cells, although the effect was most pronounced in chronic infections (Fig. 1d, Supplementary Fig. 1d). Comprehensive analysis of TCR usage from RNA-Seq data of Foxp3[+] and Foxp3[−] T cells during acute LCMV infection[30] further confirmed that the Vβ5.2-encoding *Trbv12-1* was the only transcript specifically enriched within the Treg population upon LCMV infection, while no changes were observed in the effector T cell population (Supplementary Fig. 1e). The Vβ5[+] Treg cells replenishing the regulatory compartment after LCMV infection were predominantly negative for Nrp1, a marker of thymically derived Treg cells[31,32]. Vβ8[+] Treg cells, which showed no increase, served as controls (Fig. 1e). As there has recently been some debate about the fidelity of Nrp1 as a marker for nTreg cells[33,34], we next performed a proof-of-principle experiment to determine whether bona fide induced Treg cells would be dominated by Vβ5[+] Treg cells. To this end, we adoptively transferred congenically marked CD4[+]Foxp3[−] T cells or total CD4[+] T cells into OT-II mice, in which all T cells express an ovalbumin-specific TCR and thus cannot mount an endogenous T cell response against LCMV. Upon LCMV (but not Vaccinia virus) infection we observed a robust induction of Vβ5[+] Treg cells within the transferred cells (Fig. 1f). Importantly, mice that had received CD4[+]Foxp3[−] T cells and thus only harboured iTreg cells showed an even stronger dominance of Vβ5[+] Treg cells than mice that received total CD4[+] T cells and thus harboured a mixture of nTreg and iTreg cells (Fig. 1f), confirming that the Vβ5[+] Treg pool upon LCMV infection is dominated by iTreg cells.

Next, we wanted to determine whether the relative expansion of Vβ5[+] iTreg cells was dependent on the transient reduction of Treg cells caused by the type I IFN response elicited upon LCMV infection[23,24]. Indeed, Vβ5[+] Treg cell accumulation was not observed upon infection with other acute (Vaccinia) or chronic (MCMV) viral infections or UV-inactivated LCMV that induce a less potent type I IFN response (Supplementary Fig. 2a–c)[35,36]. Furthermore, we could observe a highly significant positive correlation between the peak serum type I IFN levels elicited by different viral infections and the frequencies of Vβ5[+] iTreg cells (Fig. 1g and Supplementary Fig. 2d). We also found strongly reduced Vβ5[+] iTreg cell frequencies and numbers in mice lacking the type I IFN receptor (IFNAR, *Ifnar1*[−/−] mice; Fig. 1h). Comparing the frequency of Vβ5[+] iTregs in mesenteric LNs (mLNs) and colons of WT and *Ifnar1*[−/−] mice, we found a comparable reduction of Vβ5[+] iTreg frequencies in the colon (Supplementary Fig. 2e).

Next, we set out to determine whether type I IFN directly promoted the induction of Vβ5[+] iTreg cells or whether this was

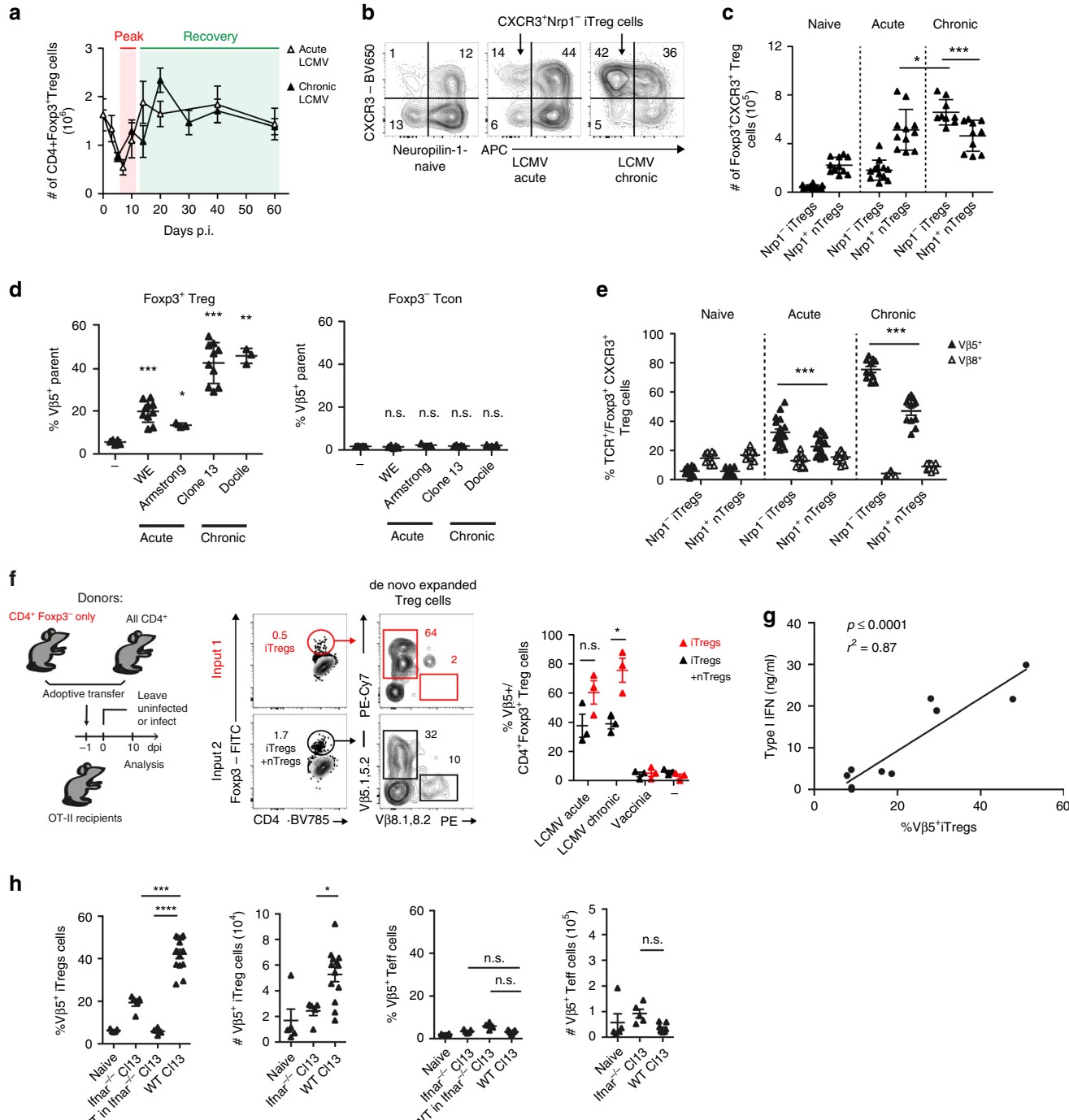

**Fig. 1 Induced Treg cells compensate for Treg cell loss caused by type I interferons. a–c** WT mice were infected with 200 f.f.u. LCMV WE (acute), $10^6$ f.f.u. LCMV clone 13 (chronic), or were left naive. **a** Total numbers of CD4+Foxp3+ splenic Treg cells were determined by flow cytometry throughout the course of infection ($n = 3$–11). Frequencies (**b**) and total numbers (**c**) of CD4+Foxp3+CXCR3+Nrp1− iTreg and CD4+Foxp3+CXCR3+Nrp1+ nTreg cells were determined in naive and LCMV-infected mice on day 14 post infection ($n = 9$–12). **d** WT mice were infected with LCMV WE (200 f.f.u.), Armstrong ($10^6$ f.f.u.), clone 13 ($10^6$ f.f.u.), or Docile ($10^6$ f.f.u.) and frequencies of CD4+Foxp3+ Treg or CD4+Foxp3− conventional T cells expressing the indicated Vβ chains were determined by flow cytometry (day 10; $n = 3$–10). **e** Frequencies of TCR Vβ5+ and Vβ8+ cells among CD4+Foxp3+CXCR3+Nrp1− iTreg and CD4+Foxp3+CXCR3+Nrp1+ nTreg cells were determined by flow cytometry as in **c** (day 14; $n = 11$–18). **f** Flow sorted CD45.1+CD4+Foxp3− T cells and CD90.1+CD4+ T cells were adoptively co-transferred into CD45.2+ OT-II mice. Reconsituted mice were infected the next day with LCMV WE (200 f.f.u., acute), clone 13 ($10^6$ f.f.u., chronic), or Vaccinia virus ($10^6$ f.f.u.) or were left naive. Vβ5+ frequencies among donor CD4+Foxp3+ were determined on day 10 ($n = 3$). **g** WT mice were infected with Vaccinia virus ($10^6$ f.f.u.), Vaccinia Virus + poly IC ($2 \times 50\,\mu g$/mouse, day 0 and 2), LCMV WE (200 f.f.u.), LCMV WE + poly IC, or LCMV clone 13 ($10^6$ f.f.u.) and type I IFN levels in the blood 24 h after infection were correlated with frequencies of Vβ5+ CD4+Foxp3+CXCR3+Nrp1− iTreg cells on day 10 post infection ($n = 2$ values per condition, pooled from 2–3 mice). **h** WT, $Ifnar1^{-/-}$, or $Ifnar1^{-/-}$ mice that had received $10^6$ Thy1.1+ CD4+ T cells i.v. one day before were chronically infected with $10^6$ f.f.u. LCMV clone 13 and frequencies and total numbers of Vβ5+ CD4+Foxp3+ Treg or Vβ5+ CD4+Foxp3− conventional T cells were determined by flow cytometry (day 10; $n = 4$–13). In summary plots, data are shown as mean ± SD. For statistics, Mann–Whitney $U$ (**c**, **e**) or one-way ANOVA (**d**, **f**, **h**) was used.

an indirect effect caused by type I IFN-dependent reduction in Treg cells upon LCMV infection[23,24] (Supplementary Fig. 2f). We found that type I IFN had no effect on the conversion of CD4+Foxp3− T cells into iTreg cells in vitro (Supplementary Fig. 2g). To confirm this in vivo, we transferred congenically marked WT CD4+ T cells into *Ifnar1*−/− mice before infecting them with LCMV. In this setting, type I IFN is produced but cannot be sensed by endogenous T cells, preventing Treg cell loss. In contrast, the transferred cells can receive a type I IFN signal and, if this directly promotes Vβ5+ iTreg induction, we expect to see increased Vβ5+ iTreg cells. However, this was not the case as we observed very low frequencies of Vβ5+ iTreg cells in this setting (Fig. 1h), confirming that type I IFN does not directly promote Vβ5+ iTreg conversion or expansion. Taken together, we found that LCMV infection results in type I IFN-dependent induction of a dominant Vβ5+ iTreg population. Importantly, type I IFN does not directly induce Vβ5+ iTreg cells but compromises the Treg niche and thereby allows for the selective enrichment of this population. We have thus identified a dominant iTreg population that drives Treg cell replenishment under inflammatory conditions as those induced by type I IFN-dominated viral infections.

**Vβ5+ Tregs are not required to control anti-viral immunity.** Given the dominance of the Vβ5+ Treg population upon LCMV infection, we next investigated its suppressive capacity and impact on the anti-viral effector response. CXCR3+Vβ5+ iTreg cells expressed normal or even slightly elevated levels of the Treg signature genes Foxp3 and CTLA-4 and the co-inhibitory receptor signature of Th1-suppressive Treg cells (Lag-3, Tim3, and CD85k[30]) when compared to total CXCR3+ Treg cells or Vβ5+ nTreg cells, respectively (Fig. 2a, Supplementary Fig. 3a–d). While PD-1 expression was decreased on CXCR3+Vβ5+ iTreg cells upon acute LCMV infection, levels in chronically infected mice were comparable and expression of TIGIT, which marks a highly suppressive Treg subset[37], was even increased (Fig. 2b, c). Furthermore, CXCR3+Vβ5+ iTreg cells strongly expressed the ectonucleotidase CD39 and the inhibitory cytokine IL-10 (Fig. 2b, c and Supplementary Fig. 3e). Furthermore, the suppressive capacity of CXCR3+Vβ5+ iTreg cells was comparable to that of other Treg subsets when tested in an in vitro suppression assay (Fig. 2d). Infection-induced Vβ5+ iTreg cells are thus fully functional and have the suppressive capacity to control effector T cells.

To elucidate the functional relevance of Vβ5+ Treg cells during LCMV infection, we next investigated the LCMV-specific immune response in mice selectively lacking Vβ5+CD4+ T cells. To this end, *Tcrb*−/−*Tcrd*−/− mice that lack all T cells were reconstituted with WT CD8+ T cells together with either total CD4+ T cells or Vβ5−CD4+ T cells (Fig. 2e). Surprisingly, the LCMV-specific immune response in animals reconstituted with Vβ5−CD4+ T cells and thus lacking Vβ5+ Treg cells was largely comparable to that observed in control mice. Stimulation with the immunodominant epitopes resulted in comparable IFN-γ production by CD4+ T cells and only slightly elevated frequencies of IFN-γ+ CD8+ T cells in mice lacking Vβ5+ CD4+ T cells (Fig. 2f). Furthermore, viral control was unaltered as viral titers in liver, kidney and spleen were comparable between the two groups (Fig. 2g). This is in contrast to mice depleted of Treg cells, which fully clear the virus (Fig. 2g). Thus, despite the fact that about half the Foxp3+ Treg cells present upon LCMV infection are Vβ5+, they do not play an essential role in controlling the anti-viral immune response.

**Vβ5+ Tregs protect from virus-induced colitis.** Although LCMV-infected animals mounted a comparable anti-viral effector response in the absence of Vβ5+ T cells (Fig. 2e–g), we noticed that they developed severe intestinal pathology that was mostly localized to the lower gastrointestinal tract ('ΔVβ5', Fig. 3a). To determine whether this was due to a loss in Vβ5+ Treg cells or Vβ5+ effector T cells that are also lacking in this setting, we reconstituted *Tcrb*−/−*Tcrd*−/− mice with WT CD8+ T cells together with Vβ5−CD4+ WT T cells plus Vβ5+CD4+ T cells from DEREG mice and selectively depleted Vβ5+ Treg but not effector T cells by diphtheria toxin administration. As in the Vβ5-deficient setting, we observed intestinal pathology upon LCMV infection ("ΔVβ5 + DEREG Vβ5+", Fig. 3a), demonstrating that the Vβ5+ Treg compartment in LCMV-infected mice safeguards the host from a colitogenic response at the intestinal mucosal barrier. The functional importance of the Vβ5+ Treg cells in this regard was further strengthened by the fact that the colitogenic response in ΔVβ5 mice could be rescued by adoptive transfer of Vβ5+Foxp3+ Treg cells isolated from LCMV-infected *Foxp3*−GFP.KI mice ("ΔVβ5 + Vβ5+ Treg AT", Fig. 3a). Importantly, this protective effect was limited to Vβ5+ Treg cells as Vβ8+Foxp3+ Treg cells were not capable of rescuing ΔVβ5 mice from colitis ("ΔVβ5 + Vβ8+ Treg AT", Fig. 3a). Interestingly, mice in which Vβ5+ T cells lack the type I IFN receptor also did not develop intestinal pathology ("ΔVβ5 + IFNAR−/− Vβ5+", Fig. 3a), confirming our previous finding that type I IFN does not directly act on Vβ5+ T cells but rather plays an indirect role by compromising the Treg niche (Fig. 1h).

Next, we set out to identify the colitogenic effectors that are regulated by Vβ5+ Treg cells upon LCMV infection. Looking at the mLN, we observed higher frequencies of IFN-γ+ CD8+ but not CD4+ T cells in ΔVβ5 mice upon PMA/ionomycin stimulation (Fig. 3b). To dermine whether CD8+ T cells might be driving the intestinal inflammation observed in ΔVβ5 mice, we next reconstituted *Tcrb*−/−*Tcrd*−/− mice with Vβ5−CD4+ WT T cells without any CD8+ T cells and infected them with LCMV. In the absence of CD8+ T cells we did not observe any inflammation ("ΔVβ5, no CD8+ T cells", Fig. 3a), confirming that CD8+ T cells drive the intestinal inflammation in ΔVβ5 mice.

We hypothesized that the enhanced CD8+ T cell response observed in the mLN (Fig. 3b) might occur in response to a barrier dysfunction caused by the infection-induced inflammatory response. We thus used an in vivo FITC-dextran feeding assay[38] to test the intestinal permeability upon LCMV infection and indeed found the barrier integrity at the gut mucosal interface to be substantially impaired in both WT and ΔVβ5 animals (Fig. 3c). This increased permeability was of functional relevance for colitis development as colitis in ΔVβ5 mice could be prevented by broad-spectrum antibiotic therapy that removed their commensal flora ("ΔVβ5 + ABX", Fig. 3a). We next tested the ability of CD4+ and CD8+ effector T cells to respond to gut-derived macromolecular antigens and microbe-associated molecular patterns (MAMP) and found that CD8+ but not CD4+ T cells responded to purified gut content with pro-inflammatory cytokine secretion (Fig. 3d and Supplementary Fig. 4). Taken together, these data reveal that increased intestinal permeability associated with viral infection triggers a colitogenic immune response that is kept in check by Vβ5+ Treg cells.

**LCMV infection induces a polyclonal Vβ5+ iTreg response.** Next we wanted to address the underlying mechanism driving the expansion of CXCR3+Vβ5+ Treg cells during LCMV challenge. First, we analyzed the clonal diversity of the Vβ5+ T cell pool by looking at CDR3 sequence diversity in *Trbv12-1*(Vβ5.2)-using CD4+ T cells at the peak of disease (day 14). CXCR3+ Treg cells from LCMV-infected animals displayed a substantial clonal overlap between different mice that was not observed in naive Treg or

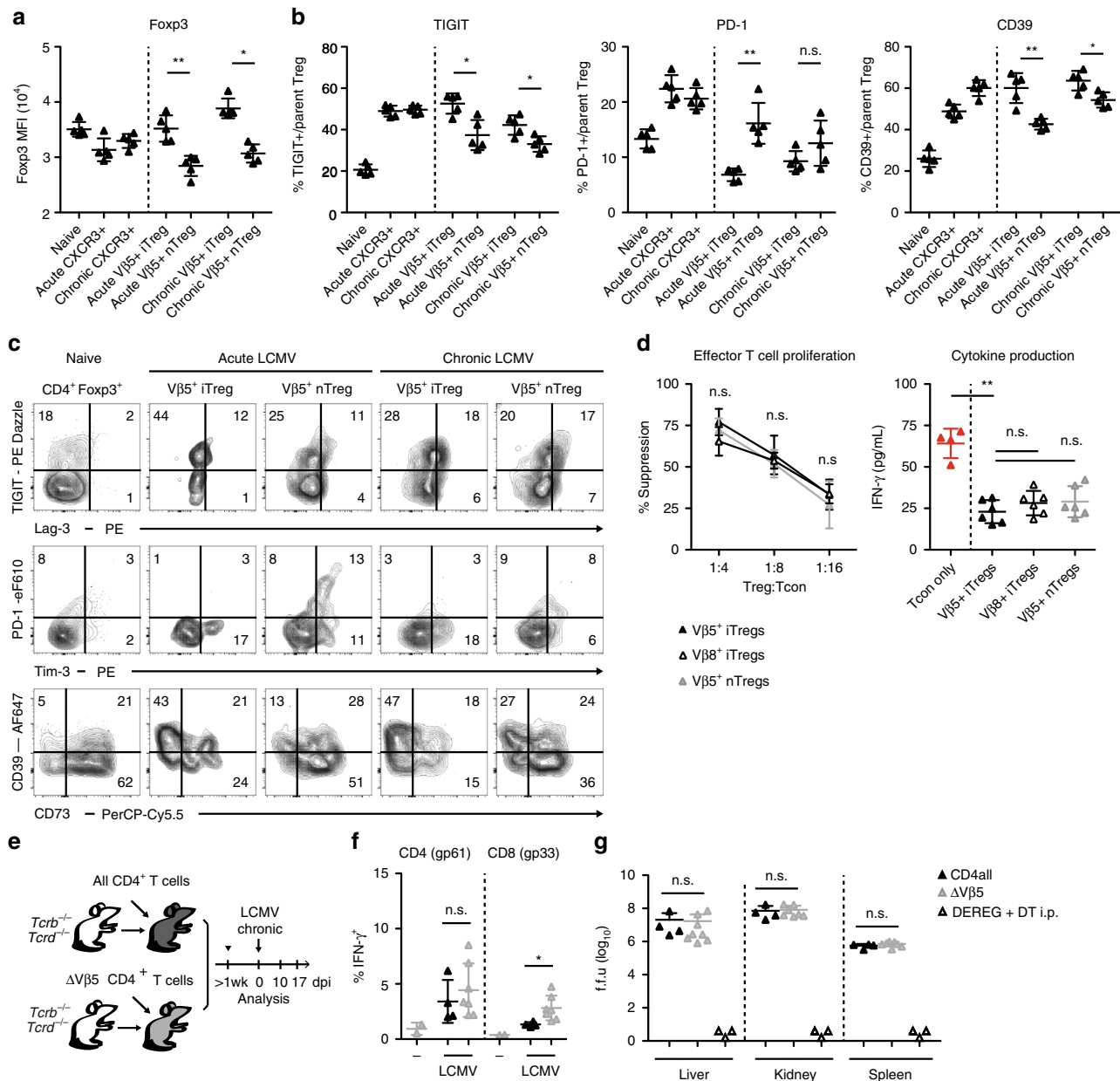

**Fig. 2 Vβ5⁺ Treg cells are suppressive but not required for regulating anti-viral immunity.** *Foxp3*-GFP.KI mice were infected with 200 f.f.u. LCMV WE (acute), 10⁶ f.f.u. LCMV clone 13 (chronic), or were left naïve. **a–c** Mean fluorescence intensity (MFI) of *Foxp3*-GFP expression and frequencies of TIGIT⁺, PD-1⁺, Lag-3⁺, Tim3⁺, CD39⁺, and CD73⁺ cells among the indicated CD4⁺Foxp3⁺ subsets were determined on day 14 post infection using flow cytometry (*n* = 5). **d** *Foxp3*-GFP.KI mice were infected with 200 f.f.u. LCMV WE and flow sorted Foxp3⁺CXCR3⁺Nrp1⁻Vβ5⁺ (filled), Foxp3⁺CXCR3⁺ Nrp1⁻Vβ8⁺ (open) or Foxp3⁺CXCR3⁺Nrp1⁺Vβ5⁺ (grey) Treg cells were titrated onto CD4⁺Foxp3⁻ effector T cells stimulated with anti-CD3 in the presence of irradiated APCs. Proliferation (left, *n* = 3 technical repeats) determined by [³H]-thymidine incorporation and IFN-γ secretion into supernatants (right, Treg:Tcon = 1:4) determined by cytometric bead array were measured after 72 h (mean ± SD, *n* = 4–6, pooled data of three independent experiments). **e** Scheme for TCR-defined CD4 T cell reconstitution. **f, g** Mice generated as outlined in **e** were infected with 10⁶ f.f.u. LCMV clone 13 (*n* = 2–7). **f** On day 17 splenocytes were restimulated with gp61 + gp33 LCMV peptides for 4 h and analyzed for IFN-γ production by flow cytometry. **g** Viral titers of the indicated organs were determined on day 17 post infection. DEREG mice depleted of Treg cells by diphtheria toxin treatment (day 4, 6, 8, 10, 12; *n* = 3) and analyzed for viral titers on day 14 were included as controls. Data are shown as mean ± SD; plots display one representative of >3 independent experiments. For statistics, Mann–Whitney *U* was used.

effector CD4⁺ T cells (Fig. 4a, left). Importantly, this high number of public clones was a unique feature of Vβ5⁺CXCR3⁺ Treg cells as *Trbv13-2*(Vβ8.1)-using Treg cells only showed a slight increase in clonal overlap upon LCMV infection (Fig. 4a, right). Nevertheless, *Trbv12-1*-using CXCR3⁺ Treg cells also harboured by far the highest number of unique clones within the entire TCR repertoire, even when compared to Foxp3⁻CD4⁺ effector T cells

(Fig. 4b). To further elucidate what drives the highly polyclonal expansion of CXCR3⁺Vβ5⁺ iTreg cells, we next focused on the mechanisms that drive their MHC class II-dependent priming. Class II ligands are mainly derived from exogenous antigens and processed via lysosomal degradation[39]. In addition, around 20% of the epitopes present on MHC II molecules originate from intracellular sources[40], which can gain access to MHC class II

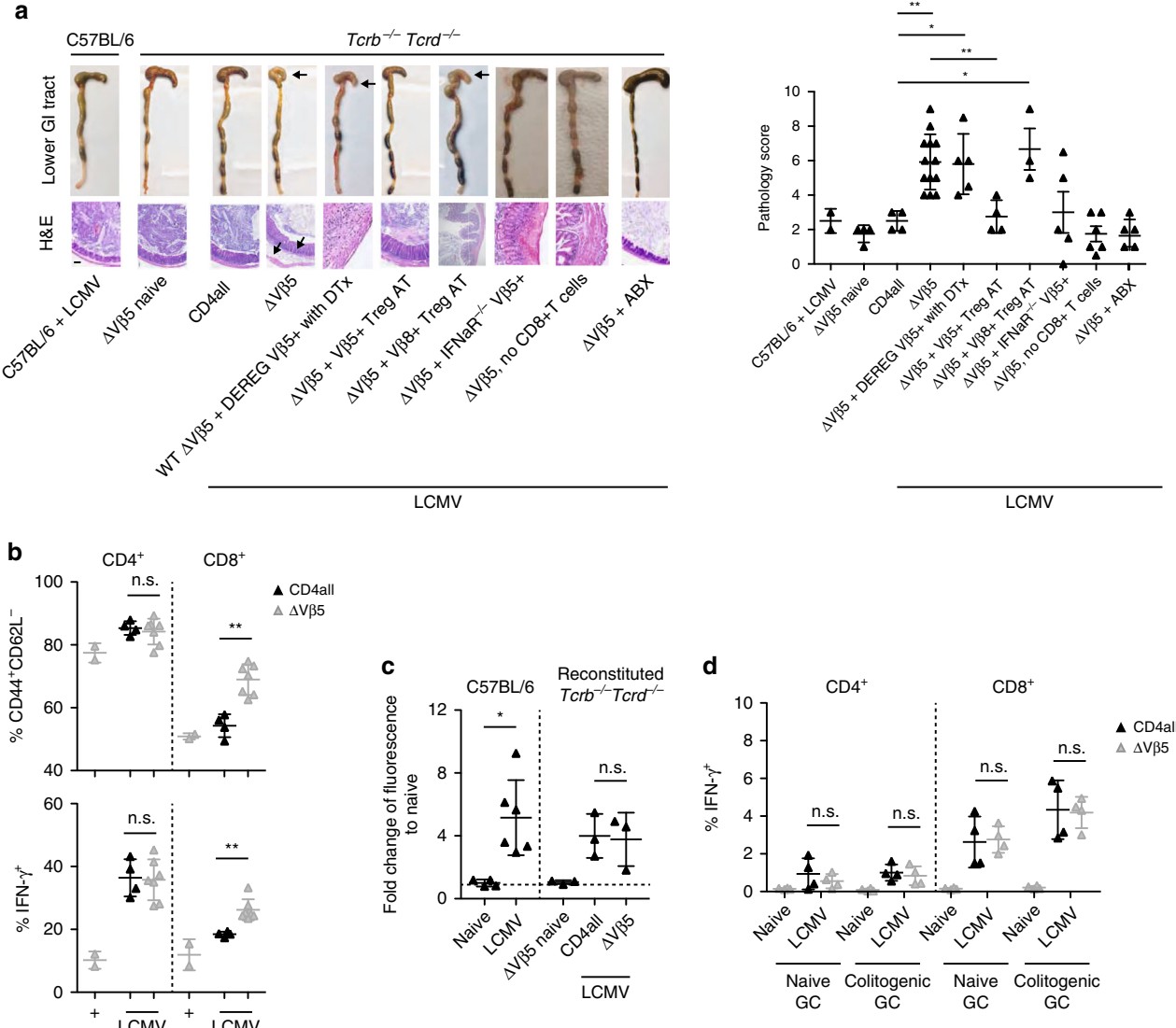

**Fig. 3 Vβ5+ Treg cells protect from virus-induced colitis.** *Tcrb−/−Tcrd−/−* mice were reconstituted with wild-type CD8+ T cells together with either total CD4+ T cells (CD4 all) or Vβ5−CD4+ T cells (ΔVβ5) and infected with 10⁶ f.f.u. LCMV clone 13 or left naïve (n = 3–13). **a** Where indicated ΔVβ5 mice additionally received Vβ5+CD4+ T cells from DEREG mice and were treated with diphtheria toxin (ΔVβ5 + DEREG Vβ5+ with DTx), Vβ5+CD4+ T cells from *Ifnar1−/−* mice (ΔVβ5 + IFNaR−/− Vβ5+), no CD8+ T cells (ΔVβ5,no CD8+ T cells), adoptively transferred CD4+Foxp3+Vβ5+ or CD4+Foxp3+Vβ8+ Tregs cells isolated from LCMV clone 13 infected *Foxp3*-GFP.KI mice 7 days after infection (ΔVβ5 + Vβ5+ Treg AT or ΔVβ5 + Vβ8+ Treg AT, respectively), or received antibiotics-supplemented drinking water 4 wks prior reconstitution and throughout the experiment (ΔVβ5 + ABX). On day 17 post infection, gastrointestinal pathology was assessed and representative pictures of the lower gastrointestinal tract (GI: cecum and colon), H&E-stained cecal tissue sections (left) as well as combined pathology scores based on loss of epithelial cells, infiltration of PMN cells, number of goblet cells and the magnitude of submucosal edema (right) are shown. Arrows indicate tissue pathology. Scale bar 100 μm. **b** Mesenteric lymph-node cells isolated on day 17 post-infection were restimulated with PMA/ionomycin for 4 h and frequencies of CD44+CD62L− as well as IFN-γ+ CD4+ and CD8+ T cells were determined by flow cytometry (n = 2–7). **c** Naïve or LCMV-infected (10⁶ f.f.u. clone 13) WT or reconstituted *Tcrb−/−Tcrd−/−* mice were fed with 15 mg FITC-dextran for 4 hrs on day 10 post-infection before FITC fluorescence was measured in serum (n = 3–6). **d** Splenocytes isolated on day 10 p.i. were restimulated with gut content for 6 h. Brefeldin A was added for the last 4 h and frequencies of IFN-γ+ CD4+ and CD8+ T cells were determined by flow cytometry (n = 4). Data are shown as mean ± SD; summary graphs display data of 2–4 independent experiments. For statistics, Mann–Whitney *U* was used.

loading compartments through autophagy protein-dependent antigen processing[41]. To investigate whether endogenous antigen might play a role in priming of CXCR3+Vβ5+ iTreg cells, we analyzed the LCMV-induced changes in the Treg compartment of *Atg5*fl/flx*Itgax*Cre/− mice, in which classical CD11c+ dendritic cells (DCs) lack Atg5, an essential protein of the autophagy machinery[42]. LCMV infection in *Atg5*fl/flx*Itgax*Cre/− mice resulted in strongly enlarged spleens (Fig. 4c) and elevated numbers of CD4+ T cells and Foxp3+ Treg cells (Fig. 4d and Supplementary Fig. 5a).

However, frequencies of virus-specific CD4+ and CD8+ effector T cells and viral clearance were comparable to those of control mice (Fig. 4e and Supplementary Fig. 5b). Interestingly, we observed a stronger induction of Vβ5+ iTreg cells in acutely infected *Atg5*fl/flx*Itgax*Cre/− mice (Fig. 4f), suggesting that exogenous antigens might be the primary driver of Vβ5+ iTreg expansion. However, we did not observe an increase in Vβ5+ iTreg priming upon chronic infection (Fig. 4c–f), possibly because the high amount of antigens present in this setting already allows for

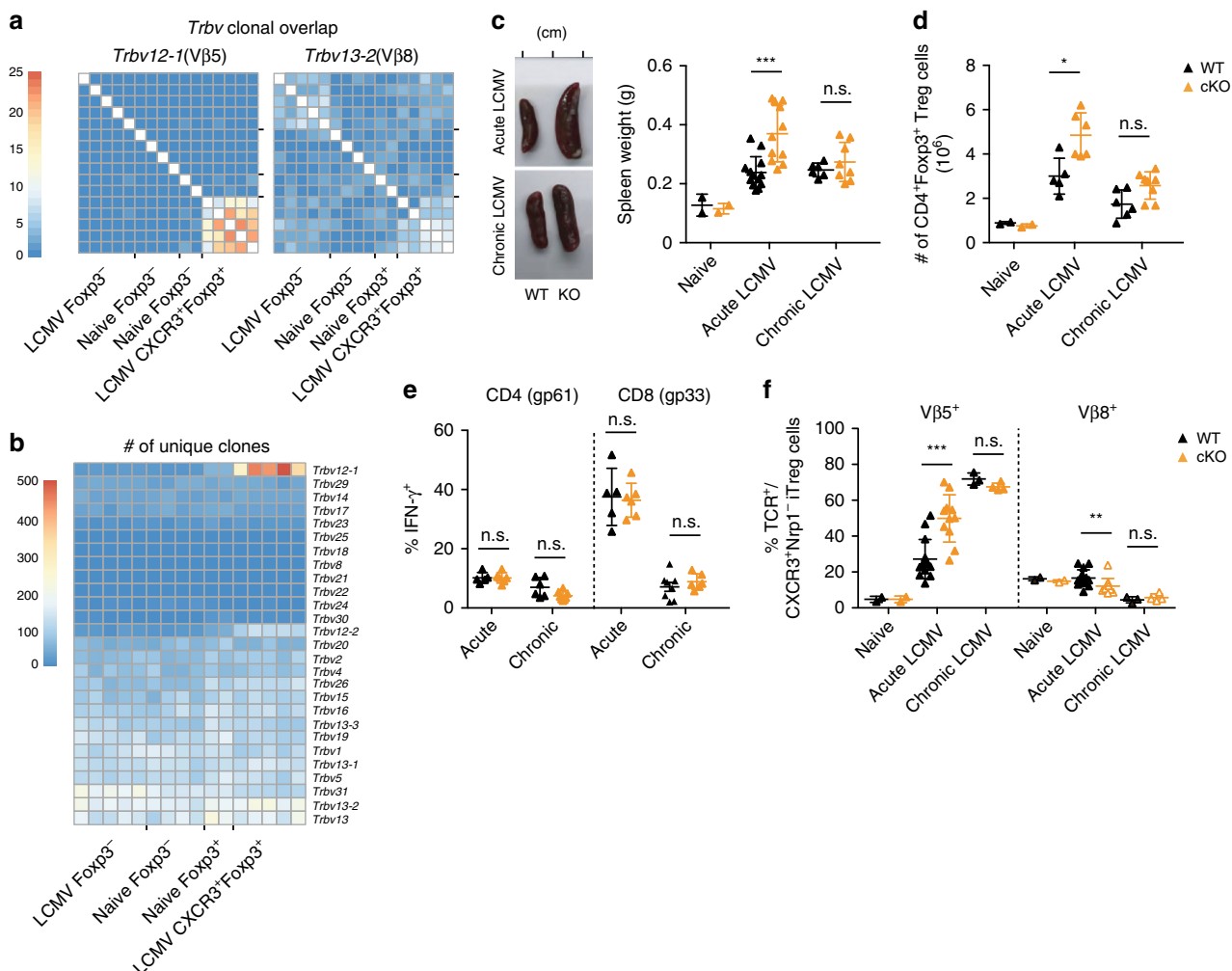

**Fig. 4 CXCR3⁺Vβ5⁺ Treg cells are polyclonal.** The number of **a** shared clones within *Trbv12-1* (left) and *Trbv13-2* (right) using CD4⁺ T cells and **b** unique TCR clones within all Vβ segments was determined based on CDR3 sequences extracted from RNA sequencing data of CD4⁺ T cell subsets from naïve or LCMV-infected (200 f.f.u. WE) mice. Scale bars refer to the number of shared clones (**a**) and the number of unique clones (**b**). **c–f** *Atg5*fl/fl x*Itgax*Cre/− mice (cKO) and *Atg5*fl/fl controls (WT) were infected with LCMV WE (200 f.f.u., acute; n = 11−13), clone 13 (10⁶ f.f.u., chronic; n = 6-8), or left naïve (n = 2). **c** Representative images (left) and quantification of dry weight (right) of spleens on day 14 after infection. **d** Total counts of splenic CD4⁺Foxp3⁺ Treg cells were determined by flow cytometry. **e** Splenocytes isolated on day 14 post infection were restimulated with gp61 + gp33 LCMV peptides for 4 h and IFN-γ⁺ frequencies were determined by flow cytometry. **f** Frequencies of Vβ5⁺ and Vβ8⁺ T cells among CD4⁺Foxp3⁺CXCR3⁺Nrp1⁻ iTreg cells were determined by flow cytometry. Data are shown as mean ± SD; summary graphs display pooled data of 2–3 independent experiments. For statistics, Mann–Whitney *U* (acute) or one-way ANOVA (chronic samples) was used.

maximal iTreg accumulation. Also, no changes in Vβ5⁺ iTreg induction were evident when the microbiota was depleted by antibiotic treatment (Supplementary Fig. 5c–e), suggesting that, unlike the colitigenic CD8⁺ T cell response, induction of Vβ5⁺ iTreg cells is not driven by microbial antigens. In summary, this data suggests that mostly exogenous antigens drive the expansion of a polyclonal Vβ5⁺ iTreg pool during LCMV infection.

**Vβ5⁺ T cells preferentially convert into Treg cells.** We next wanted to determine why Vβ5⁺ but not other CD4⁺ effector T cells predominantly converted into iTregs to compensate for the early type I IFN-dependent Treg reduction. When we enumerated total Treg cells in our model of TCR-defined reconstitution (Fig. 2e), we observed that mice lacking Vβ5⁺ CD4⁺ T cells still showed a pronounced loss of Treg cells 10 days after LCMV infection while controls had already recovered their Treg cell numbers (Fig. 5a and Supplementary Fig. 6a). Nevertheless, by day 17 Treg cell numbers in ΔVβ5 mice were

comparable to those in controls, as Treg cells carrying other Vβ chains had compensated (Fig. 5a, b, Supplementary Fig. 6a). We further observed that Vβ5⁺ Treg cells showed an almost 2-fold higher proliferation rate than other Treg populations as determined by BrdU incorporation (Fig. 5c), suggesting that Vβ5⁺CD4⁺ T cells can more rapidly expand to replenish the Treg niche. To investigate the molecular mechanisms behind their augmented capacity to fill the Treg niche, we first analyzed the strength of TCR stimulation in Vβ5⁺CXCR3⁺Nrp1⁻ iTreg and conventional CD4⁺Foxp3⁻ T cells. Nur77 (encoded by *Nr4a1*) is an early indicator of TCR signaling in lymphocytes[43,44] and we used *Nr4a1*-GFP reporter mice to analyze TCR signaling in Vβ5⁺ and control T cells upon LCMV infection. We observed an elevated level of TCR stimulation in Vβ5⁺ iTreg cells that was sustained throughout the course of infection (Fig. 5d, Supplementary Fig. 6b). Of note, already at steady-state both Vβ5⁺ iTreg and Vβ5⁺CD4⁺Foxp3⁻ conventional T cells showed a higher signal than T cells carrying other TCR β chains (Fig. 5d).

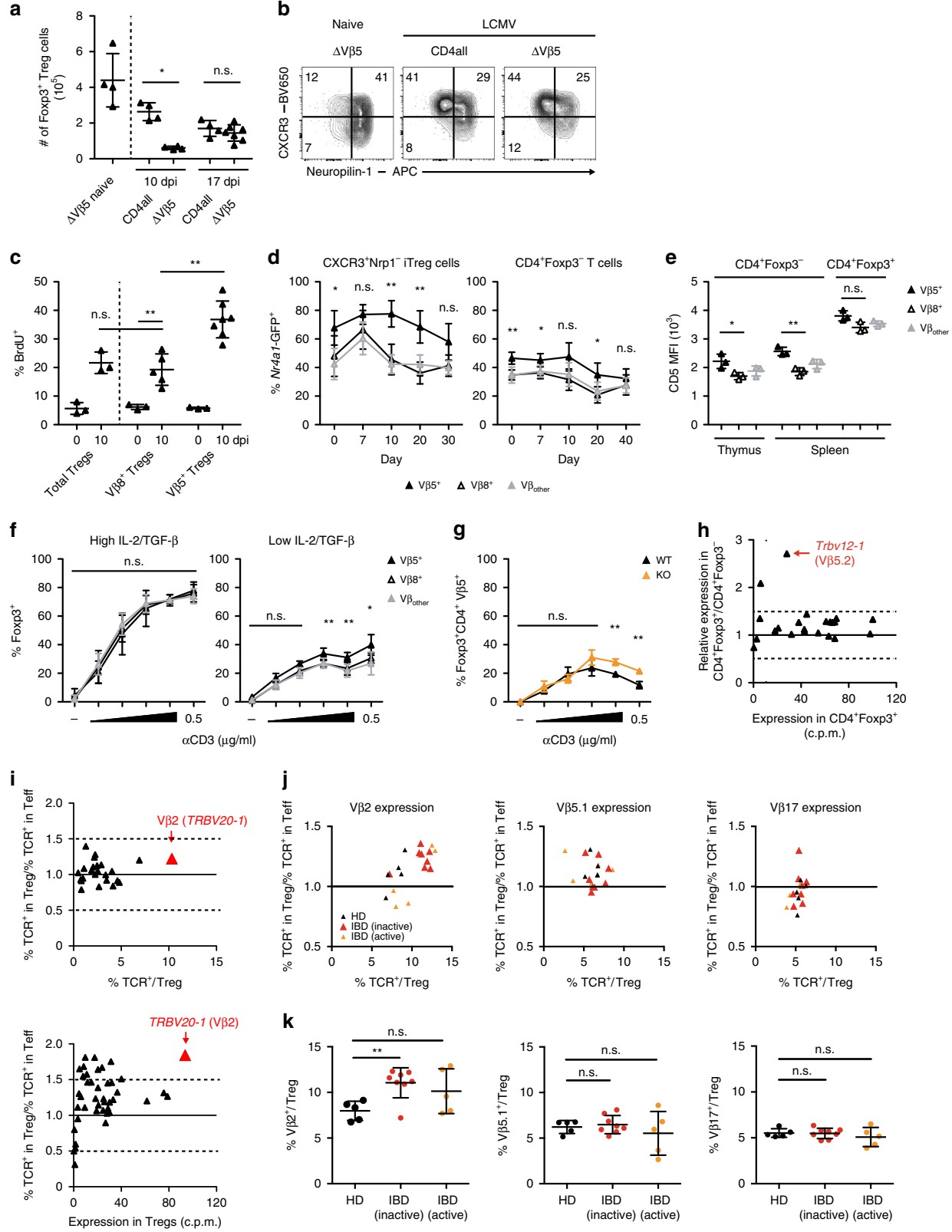

The intrinsically higher TCR signal in Vβ5⁺CD4⁺ T cells was also reflected in elevated CD5 levels in these cells (Fig. 5e), which are set during positive selection and correlate with the strength of the TCR signal received during thymic development[45,46].

We next tested whether Vβ5⁺ T cells also displayed a higher propensity to convert into Foxp3⁺ iTreg cells. Both Vβ5⁺ and

Vβ5⁻ T cells converted equally well in response to optimal stimulation with TGF-β and IL-2 in vitro (Fig. 5f, left). However, under suboptimal conditions with limiting amounts of IL-2 and TGF-β, Vβ5⁺ T cells showed superior ability to convert into iTreg cells (Fig. 5f, right). In line with our earlier findings, we also observed a higher iTreg cell conversion rate when naive Foxp3⁻

**Fig. 5 CD4$^+$Vβ5$^+$ T cells readily convert into iTreg cells.** $Tcrb^{-/-}Tcrd^{-/-}$ mice reconstituted with total or ΔVβ5 CD4$^+$ T cells were infected with LCMV clone 13 or left naïve, and analyzed 10 or 17 days after infection. Total splenic Treg counts (**a**) and their expression of CXCR3 and Nrp1 (**b**) was determined by flow cytometry ($n = 4$). **c** BrdU incorporation in splenic Treg cell populations 10 days post infection ($n = 3–7$). **d** Nr4a1-GFP mice were infected with LCMV WE and frequencies of $Nr4a1$-GFP$^+$ cells carrying the indicated TCR Vβ chain among CD4$^+$Foxp3$^+$CXCR3$^+$Nrp1$^-$ iTreg (left) and CD4$^+$Foxp3$^-$ conventional T cells (right) were determined by flow cytometry ($n = 3–6$). **e** CD5 expression in naïve thymic and splenic CD4$^+$Foxp3$^-$ conventional T cells and splenic CD4$^+$Foxp3$^+$ Treg cells was determined by flow cytometry ($n = 3$). **f** Sorted CD4$^+$CD25$^-$ T cells from naïve WT mice were co-cultured with CD11c$^+$MHCII$^+$ cells (1:1) with titrated amounts of anti-CD3 (1:3 dilution steps) and 0.3 or 3 ng/ml TGF-β plus 2 or 20 U/ml IL-2 (low and high, respectively). Foxp3$^+$ frequencies among CD4$^+$ T cells carrying the indicated TCR Vβ chain were determined by flow cytometry ($n = 3$). **g** As in **f** under low IL-2/TGF-β conditions with CD11c$^+$MHCII$^+$ cells isolated from LCMV-infected $Atg5^{fl/fl}xItgax^{Cre/WT}$ (KO) or $Atg5^{fl/f}$ (WT) mice. Frequencies of CD4$^+$Foxp3$^+$ Treg cells among CD4$^+$Vβ5$^+$ T cells ($n = 3$). **h** Transcriptional analysis of $Trbv$ gene segments shown as relative expression against expression in CD4$^+$Foxp3$^+$ Treg cells. **i** Analysis of TCR Vβ usage in healthy donor human Treg cells displayed as in **h** and determined by RNA-Sequencing (bottom) and flow cytometric spectratyping (top), ($n = 6$). **j, k** Expression of Vβ2, Vβ5.1, and Vβ17 TCR chains among CD4$^+$CD127$^-$CD25$^{hi}$ Treg and CD4$^+$CD127$^{var}$CD25$^{var}$ effector T cells from healthy human donors ($n = 5$) or IBD patients with inactive ($n = 8$) or active ($n = 5$) disease was determined by flow cytometry. Data are presented as relative expression vs. expression in Treg cells (**j**) and as frequencies within Treg cells (**k**). Data are shown as mean ± SD; summary graphs display pooled data of 2–3 independent experiments. For statistics, Mann–Whitney $U$ (**a, c, e, k**) or one-way ANOVA (**d, f, g**) was used.

T cells were co-cultured with $Atg5$-deficient CD11c$^+$MHC II$^+$ DCs isolated from LCMV-infected mice than with DCs from WT mice (Fig. 5g), further supporting the notion that an exogenous antigen is driving conversion of Vβ5$^+$Foxp3$^-$ T cells into iTreg cells. Strikingly, a substantial over-representation of Vβ5.2$^+$ cells within the Treg compared to the conventional T cell compartment was already apparent at steady-state (Fig. 5h), supporting the notion that Vβ5$^+$ T cells preferentially convert into Treg cells in vivo even in the absence of LCMV infection. Importantly, when analyzing human T cells, we also found a specific over-representation of a single Vβ chain, Vβ2 ($TCRBV20-1$), within the Treg compartment (Fig. 5i). Like Vβ5$^+$ Treg cells in mice, the human Vβ2$^+$ Treg population was highly polyclonal (Supplementary Fig. 7). More importantly, Vβ2$^+$ Treg cells, but not T cells carrying other abundant Vβ chains, were specifically expanded in IBD patients with inactive disease (Fig. 5j, k), suggesting that recovery from clinical symptoms is marked by high frequencies of Vβ2$^+$ Treg cells. Taken together, these findings show that effector T cells carrying specific TCR β chains harbour an intrinsically higher potential to convert into Treg cells allowing for the rapid replenishment of the Treg niche if it is compromised by a virus-induced type I IFN response upon infection (Fig. 6).

## Discussion

In this study, we uncovered an essential role for iTreg cells expressing specific Vβ chains in preventing autoimmunity upon viral triggers. LCMV infection induces a type I IFN-dependent loss of Treg cells that is rapidly compensated by the conversion of high-affinity Vβ5$^+$ conventional T cells into iTreg cells. While these Vβ5$^+$ iTreg cells do not play a role in limiting the anti-viral immune response, they are essential for suppression of colitogenic CD8$^+$ effector T cells at the gut barrier site. In humans, this role is fulfilled by Vβ2$^+$ Treg cells, which are specifically expanded in IBD patients with inactive disease.

Treg cells are essential for the peripheral control of autoreactive T cells as their loss results in autoimmunity[47]. Predisposing genetic factors contribute to the manifestation of such autoimmune conditions but in most cases additional environmental triggers, such as viral infections, are required to initiate the onset of disease[8,48]. However, the underlying mechanisms triggered by the infectious challenge are still largely unclear. In this study, we reveal that the anti-viral type I IFN response strongly compromises the Treg compartment. This allows for the primimg of colitogenic CD8$^+$ T cells at the gut barrier site, resulting in

microbiota-dependent activation of CD8$^+$ T cells and colitis. In line with these results, Norovirus infection has been associated with exacerbations and flares of IBD[49,50]. Our data would suggest that these flares are the result of a transient reduction in Treg cells that is compensated by the expansion and conversion of Vβ2$^+$ T cells into Treg cells as observed in the IBD pateints with inactive disease.

We further show that a rapid replenishment of the Treg pool through enhanced conversion and expansion of a specific CD4$^+$ T cell population into Foxp3$^+$ iTreg cells is necessary to prevent the onset of pathologic T cell responses. Using LCMV infection as a model, we establish that non-microbiota specific Vβ5$^+$ T cells are selected based on an intrinsically high TCR signal and their superior ability to convert into iTreg cells, enabling them to rapidly replenish the Treg niche upon LCMV infection. The highly polyclonal nature of the Vβ5$^+$ iTreg response might suggest that a global stimulus is involved. Indeed, earlier reports have shown that LCMV infection results in the reactivation of endogenous mouse mammary tumor virus (MMTV)-encoded superantigens that can expand Vβ5$^+$CD4$^+$ nTreg cells[51]. Furthermore, the fact that CXCR3$^+$Vβ5$^+$ Treg cells harbour a relatively high number of public clones is consistent with superantigen-mediated activation[52]. As such, it is likely that the polyclonal conversion and expansion of Vβ5$^+$CD4$^+$ T cells into iTreg cells is in part driven by these superantigens.

Like Vβ5$^+$ T cells in mice, we found human T cells carrying the Vβ2 chain to be highly overrepresented within the Treg pool when compared to effector T cells, suggesting that they harbour the same abilty as mouse Vβ5$^+$ T cells to rapidly replenish a compromised Treg niche. Indeed, Vβ2 was recently reported to be overrepresented in the colon of patients with ulcerative colitis or Crohn's disease[53]. This is in line with our observation that IBD patients with inactive disease have a higher proportion of Vβ2$^+$ Treg cells that likely expanded to restore a compromised Treg compartment.

High levels of type I IFN present during anti-viral responses compromise the regulatory compartment[23,24], but this can be compensated by the rapid induction and expansion of iTreg cells. However, if this process is impaired, such as in Vβ5-deficient mice as shown in this study, the compromised Treg compartment is no longer able to limit autoreactive immune responses. Prolonged exposure to type I IFN, such as in IFN therapy or in patients with type I interferonopathies that are characterized by the constitutive production of type I IFNs, likely also impairs or even prevents Treg cell replenishment, which then manifests in inflammatory responses directed at self[15,54]. The fact that type I

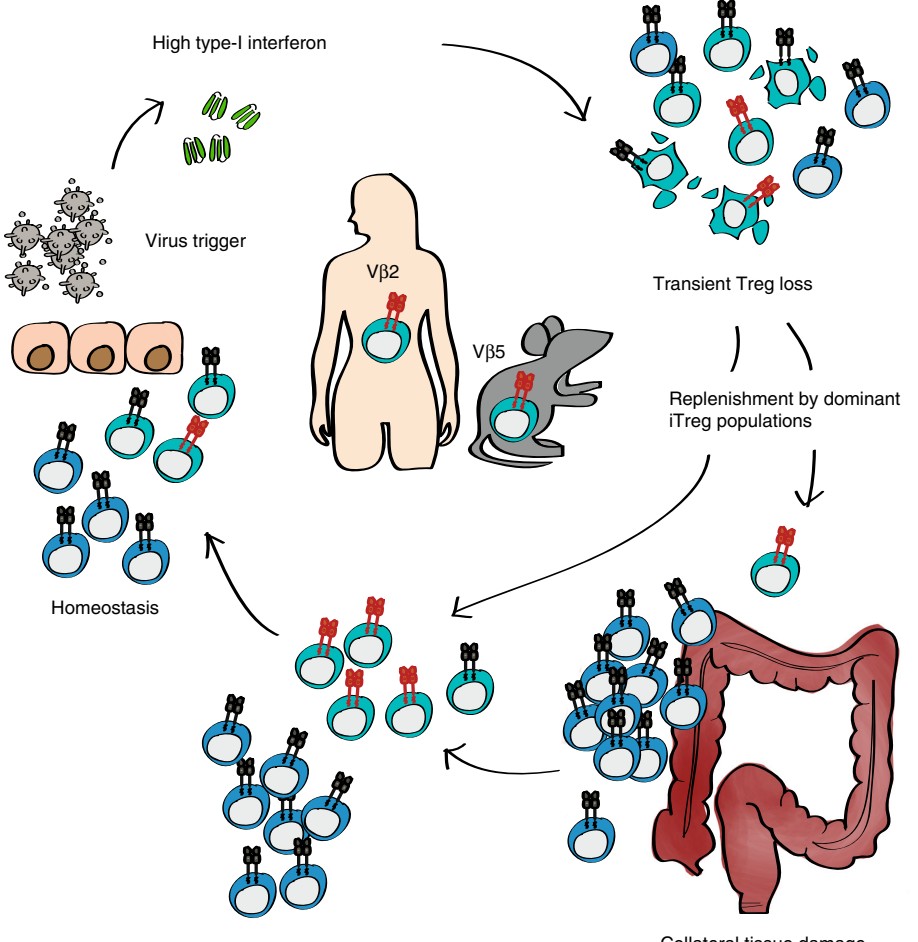

**Fig. 6 Schematic representation of virus-induced alterations in Treg cells and their impact on colitis.** Virus-induced type I IFN responses transiently impair the Treg cell compartment. The Treg pool is rapidly replenished by iTreg cells converted from a high-affinity CD4+ T cell population (Vβ5+ CD4+ T cells in mice and Vβ2+ CD4+ T cells in humans) and returns to homeostasis. If this process is impaired, the lack of Treg-mediated suppression allows for a colitogenic immune response to be triggered, leading to intestinal immune pathology.

IFNs play a role in induction or exacerbation of e.g. SLE or RA[19] and that IFN blocking strategies have shown promising results in SLE patients in clinical trials further supports this hypothesis[55]. Based on our results, we propose that a "division of labour" exists between nTreg cells and Treg cells induced in the periphery. Thymus-derived nTregs play a fundamental role in maintaining self-tolerance and immune homeostasis under steady-state conditions. However, upon an infectious challenge, the inflammatory anti-pathogen response results in an initial depletion of IFN-sensitive nTregs to allow for the initiation of a potent adaptive immune response and pathogen clearance. However, this impairment in immune control has to be transient in its nature to ensure that activation of autoreactive cells in the periphery is prevented. This requires a rapid replenishment of the Treg pool, which can be achieved through expansion and conversion of high-affinity conventional T cells into iTreg cells. If this response is compromised or other predisposing factors exist, the infection-triggered transient reduction in Treg cells might serve as an environmental trigger for autoimmunity.

Overall, our results support the concept that the Treg compartment as a whole is able to adapt to inflammatory insults that may compromise peripheral tolerance. We show that virus-induced type I IFN responses transiently impair the Treg cell compartment, but that this is rapidly replenished by iTreg cells

converted from a high-affinity CD4+ T cell pool to prevent inflammatory disorders and overt autoimmunity.

## Methods

**Animals and infections.** Animal experiments were reviewed and approved by the cantonal veterinary office of Zurich (permit numbers ZH100/2014, ZH168/2015, ZH114/2017, ZH119/2017, and ZH179/2017) and were performed in accordance with Swiss legislation. All mice were bred and housed at the ETH Phenomics Center Zurich, Switzerland (C57BL/6-Tg(*Nr4a1*-EGFP/cre)820Khog/J mice) and the Laboratory Animal Sciences Center (LASC) Zurich, Switzerland (all other strains). C57BL/6Rj (WT) mice were purchased from Janvier Labs. *Foxp3*-GFP.KI reporter mice[56], B6.Cg-Tg(*TcraTcrb*)425Cbn/J (OT-II; JAX 004194[57]) mice, B6.Cg-*Gpi1*a*Thy1*a*Igh*a/J (Thy1.1; JAX 001317[58]), *Ifnar1*tm1Agt mice (MMRC JAX 32045[59]), *Tcrb*tm1Mom*Tcrd*tm1Mom mice (JAX 002122[60]), and C57BL/6-Tg(*Nr4a1*-EGFP/cre)820Khog/J mice (JAX 016617[43]) were described earlier. B6.SJL-*Ptprc*a*Pepc*b/BoyJ (Ly5.1; JAX 002014) and *Foxp3*-GFP.KI mice were crossed to generate *Foxp3*-GFP.KILy5.1 reporter mice. *Atg5*flox/flox [61] and C57BL/6J-Tg(*Itgax*-cre,-EGFP)4097Ach/J mice (JAX 007567[62]) were crossed to obtain *Atg5*fl/flx*Itgax*Cre/− mice; *Atg5*flox/flox and *Atg5*fl/WTx*Itgax*Cre/− mice mice were used as littermate controls. All mice were on the C57BL/6 background, used at 7–22 weeks of age, and age and sex matched within experiments.

LCMV strains Armstrong, WE, clone 13 and Docile were propagated on L929 fibroblasts, Vaccinia virus on BSC40, and MCMV on M2-10B4 cells. Mice were infected intravenously (i.v.) with 200 f.f.u. LCMV for low dose infection or with 10^6 f.f.u. for high-dose infection, with 2 × 10^5 p.f.u. MCMV, or intraperitoneally (i.p.) with 2 × 10^6 p.f.u. Vaccinia virus. Non-replicating LCMV clone 13 was generated by exposure to UV light for 60 min at RT. When indicated, animals were

pulsed with 1 mg BrdU (Biolegend) i.p. 24 h before sacrifice. Broad-spectrum antibiotic therapy using ampicillin, vancomycin, neomycin, and metronidazol (Sigma) was applied in drinking water as described[63]. To measure gut permeability, animals were deprived of food and drink for 3 h and then fed with 15 mg 4 kDa FITC-dextran (TdB Cons) as described[38]. Serum was collected and analyzed for fluorescence 4 h later by Spectramax i3 (Molecular Devices). For assessment of colitis, organs were fixed in PFA, and paraffin-embedded cecal sections were visualized by hematoxylin-eosin staining (Veterinary Pathology, University of Zurich, Switzerland). Histopathological scoring was determined as previously described[64]. Briefly, sections were scored for (i) submucosal edema, (ii) PMN infiltration into the lamina propria, (iii) goblet cell loss, and (iv) epithelial integrity and were given a score from 0 to 3 (0 = no pathological changes; 1 = mild changes; 2 = moderate changes; 3 = profound changes).

**Human samples.** Peripheral blood was collected from healthy donors (approved by Cantonal Ethics Committee of Zurich, BASEC number 2016-01440) and IBD patients with active (Colitis activity index > 6) or inactive (Colitis activity index < 3) disease that were neither on anti-TNFα nor on steroid therapy (approved by Cantonal Ethics Committee of Zurich, EK-1316) and informed consent was obtained prior to blood collection. For T cell sequencing, T cells were sorted from peripheral blood of 6 age-matched healthy Caucasian male donors. Furthermore, whole blood from healthy donors and IBD patients was collected and directly analyzed by flow cytometry.

**Flow cytometry and T cell stimulation.** Single cell suspensions from the indicated organs were used directly for extracellular staining (45 min at 4 °C) or restimulated ex vivo. For restimulation, isolated splenocytes were incubated for 4 h at 37 °C in the presence of PMA and ionomycin (Sigma) or the LCMV-immunodominant peptides gp61 and gp33 (EMC Microcollections), together with brefeldin A (Biolegend). Gut contents (GC) were homogenized in PBS using metal balls and a TissueLyser II (25 Hz, 3 min; Qiagen) and sedimented for 10 min at 8000 g at 4 °C. Sterile filtered GC (0.22 µm unit, Merck Millipore) were mixed at a 1:5 vol ratio with isolated splenocytes and incubated for 2 h at 37 °C, before brefeldin A was added for an additional 4 h. For transcription factor staining the Foxp3 Staining Buffer Set (Thermo Fisher) and for intracellular cytokine staining the BD Fixation/Permeabilization Solution kits (BD Bioscience) were used according to the manufacturer's instructions.

All fluorescently labeled antibodies—for murine samples: CD4 (RM4-5 or GK1.5), CD8 (53-6.7), CXCR3 (CXCR3-173), Nrp1 (3E12), Foxp3 (FJK-16s), TCR Vβ5.1,5.2 (MR9-4), Vβ8.1,8.2 (KJ16-133.18), TCR γδ (GL3/UC7-13D5), Vβ2 (B20.6), Vβ6 (RR4-7), Vβ7 (TR310), Vβ8.3 (1B3.3), Vβ10b (B21,5), Vβ11 (RR3-15/KT11), Vβ12 (MR11-1), and Vβ13 (MR12-4), CD44 (IM7), CD62L (MEL-14), TIGIT (1G9), LAG3 (C9B7W), PD-1 (J43), CD39 (Duha59), CD73 (TY/11.8), CD85k (H1.1), BrdU (Bu20a), IFN-γ (XMG1.2), CD5 (53-7.3), CD11c (N418), MHCII (M5/114.15.2), CD45.1 (A20), CD90.1 (OX-7); for human samples: CD3 (OKT3), CD4 (SK3), CD8 (SK1), CD19 (HIB19), CD25 (M-A251), CD45RA (HI100), and CD127 (A019D5)—were purchased from Biolegend, eBioscience, or BD Biosciences. For T cell spectratyping from human PBMCs, the IOTest Beta Mark TCR Vβ Repertoire Kit was used (Beckman Coulter). To detect live cells, the Zombie-NIR fixable dye (for murine cells) and 7-AAD (for human cells) were used. LSRFortessa or FACSCanto II cytometers were used for cell acquisition and a FACS Aria III 5L was used for cell sorting (all BD Bioscience; Core Facility, University of Zurich, Switzerland). Data analysis was carried out using the Flowjo software (Treestar).

**Cell sorting and adoptive cell transfers.** Single cell suspensions of splenic T cells were obtained by mechanic disruption. For obtaining high purity APCs, spleens were additonally subjected to enzymatic digestion by collagenase D (Gibco) and DNase I (VWR) for 45 min at 37 °C. Prior to cell sorting, cells were enriched by magnetic cell sorting (MACS) using anti-CD4, anti-CD11c, or Pan T cell MojoSort beads (Biolegend). Wild-type or congenically marked cells were sorted as CD8+ T cells, CD4+ T cells, CD4+Vβ5.1,5.2− T cells, CD4+Foxp3+Vβ5.1,5.2+ Treg cells, CD4+CD25− T cells, CD4+Foxp3− T cells, or CD11c+MHCII+ APCs. For T cell reconstitution, $1.8 \times 10^6$ CD8+ and $2 \times 10^6$ CD4+ T cells were co-injected i.v. into *Tcrb*tm1Mom*Tcrd*tm1Mom recipient mice, and allowed to reconstitute for 1 or >7 days before LCMV challenge as indicated.

**Type I bioassay.** 20'000 LL171 cells/well were grown in a 96 well flat bottom luciferase reading plate over night in 10% RPMI medium supplemented 1% L-glutamine (all Gibco). On the next day, 10-fold pre-diluted test sera and respective IFN-β standard (Biolegend) were added in RPMI medium and incubated for 24 h at 37 °C. The cells were then lysed in 1x lysis buffer (Promega) for 10 min, followed by the addition of luciferase substrate (Promega) via injector system. The luciferase activity was measured on a Tecan Infinite 200 Pro plate reader.

**In vitro Treg cell assays.** T cells were cultured in RPMI medium supplemented with 10% heat-inactivated FCS, 50 mM β-mercaptoethanol, 1 mM sodium pyruvate, nonessential amino acids, MEM vitamins, 50 U/mL penicillin, 50 µg/mL strepto-mycin, 2 mM glutamine (all from Gibco), and 50 µg/mL gentamicin (Sigma) and Treg cell suppression assays were conducted as previously described[30]. Briefly, sorted CD4+Foxp3− conventional T cells ($4 \times 10^4$/well) were co-cultured in triplicate with the indicated CD4+CXCR3+Foxp3+ Treg subsets in the presence of soluble anti-CD3 (1 µg/ml, BioXcell) and irradiated splenic APCs ($2 \times 10^5$/well) for 48 h at 37 °C. Cells were then pulsed with 1 µCi [³H]thymidine (PerkinElmer) for an additional 18–22 h and thymidine incorporation was assessed to determine effector T cell proliferation. Secreted IFN-γ levels were determined in supernatants using the LEGENDplex cytometric bead array (Biolegend). For in vitro induction of Treg cells, sorted CD4+Foxp3− T cells and CD11c+MHCII+ APCs were cultured for 72 h at 37 °C in the presence of soluble anti-CD3 (5 ng–0.5 µg/ml, BioXcell), recombinant human TGF-β1 (0.3 or 3 ng/ml, RayBio), recombinant mouse IL-2 (2 or 20 U/mL), anti-IL-4 (1 µg/ml, 11B11), and anti-IFN-γ (1 µg/ml, XMG1.2; all from Biolegend). Alternatively, T cells were stimulated with plate bound anti-CD3 (0.5 µg/ml) and anti-CD28 (2 µg/ml), TGF-β1 (3 ng/ml), IL-2 (20 /mL), anti-IL-4 (1 µg/ml, 11B11), and anti-IFN-γ (1 µg/ml, XMG1.2) and type I IFN (3000U/ml) where indicated. Acquired expression of Foxp3 was subsequently assessed by flow cytometry.

**Foci forming assays.** Viral titers in infectious inoculum and in indicated organs were determined as described previously[65]. In brief, homogenized samples were serially diluted (10-fold) onto MC57G cells seeded into 24-well plates ($10^5$/well) in MEM supplemented with 5% FCS and 1% PSG (Corning and Gibco). Infected cells were overlayed 4 h later with a 1:1 mix of 2% methylcellulose (Sigma) and 5% FCS/DMEM medium (Gibco), and incubated for an additional 48 h at 37 °C. Cells were fixed with 4% PFA, permeabilized with 1% Triton X-100, and stained with rat anti-LCMV VL-4 and peroxidase-conjugated goat anti-rat IgG (Jackson ImmunoResearch). For visualization OPD (Sigma) substrate was applied for >30 min at RT.

**RNA-seq data and analysis.** For mouse samples, CD4+ T cells were pre-purified from splenocytes and LNs of naive or LCMV WE infected (day 14) *Foxp3*-GFP.KI reporter mice using anti-CD4 beads (Miltenyi). CD4+Foxp3− conventional T cells and CD4+Foxp3+, CD4+Foxp3+CXCR3+ or CD4+Foxp3+CXCR3− Treg cells were sorted by flow cytometry. RNA was extracted using the Qiagen RNeasy Micro Kit and libraries were prepared and sequenced by the Functional Genomics Center Zurich (Zurich, Switzerland). Human PBMCs were isolated by Ficoll-Paque (GE Healthcare) densitiy centrifugation and 2000 cells were sorted as 7-AAD−CD3+CD4+CD127−CD25high Treg cells and 7-AAD−CD3+CD4+CD127varCD25var effector T cells from human PBMCs (see Supplementary Fig. 6a for gating strategy). For RNA isolation, the Smart-Seq2 protocol was used as described in ref. [66]. Briefly, Agencourt RNAClean XP paramagnetic beads (Beckman Coulter) were used in combination with a DynaMag-96 side skirted magnet (Thermo Fisher). cDNA was generated with SuperScript II Reverse Transcriptase Kit (Thermo Fisher) and amplified with HiFi HotStart PCR Mix (KAPA Biosystems). For DNA clean-up Agencourt AMPure XP beads were used (Beckman Coulter) as above. Nextera XT DNA sample preparation and index kits (Illumina) were used for preparation of libraries that were sequenced by the Functional Genomics Center Zurich (Zurich, Switzerland). RNA-Seq reads of the Treg murine samples from this experiment were already published previously[30]. RNA-Seq reads were aligned to the mouse (Ensembl_GRCm38.75) or human reference genome (Ensembl_GRCh38) using Subread (v1.6.2) using only uniquely mapping reads[67]. Both mouse and human expression levels were quantified at the gene-level using the featureCounts function of the Rsubread package[68] with NCBI Entrez IDs. Normalized counts for the TRBV genes were quantified using the R package edgeR[69]. Repertoire analysis from RNA-seq data was performed using MiXCR software (version 3.0.3) allowing for partial alignments and performing two rounds of subsequent contig assembly. Clonotypes were assembled using the CDR3 feature and exported using default parameters[70].

**Statistical evaluation.** Statistical significance was determined using Mann–Whitney $U$ or one-way ANOVA with Tukey's multiple comparison post tests (GraphPad Prism 6). Significance is indicated as $p \geq 0.05$ (not significant, n.s.), $p < 0.05$ (*), $p < 0.01$ (**) or $p < 0.001$ (***).

**Reporting summary.** Further information on research design is available in the Nature Research Reporting Summary linked to this article.

## Data availability
Source data for all figures and Supplementary Figures are provided with the paper. RNA-Seq data that support the findings of this study have been deposited at the ArrayExpress database at EMBL-EBI (www.ebi.ac.uk/arrayexpress) and are available via the accession numbers E-MTAB-8861 (mouse T cells) and MTAB-8819 (human T cells). Within E-MTAB-8861, T-conventional cell sequences are reported in this study for the first time, while Treg sequences from the same experiment have been also described in our earlier publication[30] and shared as the dataset E-MTAB-6156.

## Code availability
All code used in this study has been decribed previously. Subread (http://subread.sourceforge.net), edgeR (https://bioconductor.org/packages/release/bioc/html/edgeR.html), and MiXCR (https://github.com/milaboratory/mixcr/releases) are available via the indicated links.

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

## Acknowledgements

We would like to thank Maries van den Broek for vaccinia virus, Marcus Goettrup for mice, Burkhard Becher for antibodies, Tamas Dolowschiak for experimental support, and Monique Gannagé, Natacha Madelon, and members of the Joller and Oxenius group for discussions. This work was supported by the Swiss National Science Foundation (PP00P3_150663 and PP00P3_181037 to N.J.; 310030-146140 and 310030-166078 to A.O.; 310030-172978 to O.B.), the European Research Council (677200 Immune Regulation to N.J.), the Zuercher Universitaetsverein (ZUNIV-FAN to N.J.), the Olga Mayenfisch Stiftung (to N.J.), the Novartis Foundation for medical-biological research (17A027 to N.J.), the Swiss Federal Institute of Technology (to A.O., ETH-44 14-2 to R.S.), the University of Zurich Forschungskredit, and the Clinical Research Priority Program of the University of Zurich for the CRPP CYTIMM-Z (to O.B.).

## Author contributions

Conceptualization: M.S., N.R., R.S., A.O., N.J.; methodology: M.S., A.Y., K.L., N.R.; validation: M.S., K.L., N.R.; formal analysis: A.Y.; investigation: M.S., K.L., N.R., F.R., K.C.K, J.O.; resources: C.W.K., M.E.R., J.D.L., G.R., O.B.; writing—original draft: N.J.; writing—review and editing: all authors; visualization: M.S., A.Y.; supervision: N.J., A.O.; project administration: N.J.; funding acquisition: R.S., A.O., N.J.

## Competing interests

The authors declare no competing interests.
