## [Peer Review File · Nature Communications]

Editorial Note: Parts of this Peer Review File has been redacted as to indicated to maintain the confidentiality of unpublished data.

Reviewers' comments:

Reviewer #1, expert on LCMV and T cells (Remarks to the Author):

In this study, using acute and chronic Lymphocytic choriomeningitis virus (LCMV) infection models, Dolowschiak et al. have investigated how Treg-mediated peripheral tolerance is maintained during viral infections that induce a strong type I IFN response. In line with previous reports, the authors determine that acute and chronic LCMV infection results in an early and rapid reduction in Treg cell abundance. Here the authors additionally determine that at the recovery phase of the infection, the loss of nTregs was predominantly replenished with CXCR3+Nrp1- induced Treg (iTregs), especially during chronic LCMV infection. Using RNA-seq and flow cytometry, the authors demonstrate an enrichment for V β 5.1, 5.2+ iTreg cells compared to other TCR V β -defined Treg subsets in chronic infection. Interestingly, the authors identified that this preferential outgrowth of V β 5+ Tregs does not appear to occur during vaccination or other types of acute or chronic viral infections, which (purportedly) elicit lower levels of type I IFN production. Using adoptive transfer experiments, the authors demonstrate that iTregs can develop from FoxP3- CD4 T cells during LCMV infection, and that most of these iTregs express Vb5. Additionally, using IFNAR-/- mice, the authors show that the proportion of V β 5+ iTregs is modestly reduced in the absence of Type 1 IFN signaling during chronic viral infection. Phenotypic profiling and functional characterization of V β 5+ iTregs revealed that these cells express equivalent or slightly higher levels of Foxp3 and other inhibitory molecules, and are equally capable of suppressing T cell proliferation and cytokine production in vitro as compared to other Treg subsets.

To further understand the functional relevance of this Treg subset, the authors' reconstituted Tcrb-/- Tcrd-/- mice with either total CD4 T cells or Vb5- CD4 T cells. Using this system that allows for specific deficiency of V β 5+ nTreg and iTregs, they surprisingly uncovered that V β 5+ nTregs and iTregs do not appear to play a major role in regulating anti-viral immunity following LCMV infection. However, mice deficient of V β 5+ nTregs and iTregs started to develop severe intestinal pathology, and this colitis could be rescued by adoptive transfer of V β 5+Foxp3+Tregs. Using an in vivo FITC-dextran feeding assay, the authors show that virus induced inflammatory response promoted barrier dysfunction, which they conclude is likely mediated by non-LCMV specific T cells. To explore how antigen presentation impacts iTreg induction, the authors used genetic mice specifically deficient in processing and presenting intracellular antigens. Following LCMV infection these mice displayed enlarged spleens but comparable viral clearance and anti-viral effector responses. Interestingly, the authors still observed a strong induction of V β 5+ iTreg cells when autophagy-dependent antigen-processing was abrogated, suggesting exogenous antigens could be the primary driver for V β 5+ iTreg expansion. In terms of understanding the capacity of V β 5+ CD4 T cells to preferentially give rise to the iTreg subset, the authors used Nr4a1-GFP reporter mice and found V β 5+ T cells possess a higher potential to convert into iTreg with their increased intrinsic TCR signal strength and proliferative capacity following LCMV infection. The authors have also identified V β 2+ T cells as the corresponding population in humans. Although some of the data presented is novel and interesting, there are several major concerns that need to be addressed.

Major concerns:

Given that a major premise of this work is based on viral-induced type I IFN-mediated induction of iTregs, and their subsequent role in preventing autoimmunity, this assertion, and the biological relevance iTreg expansion during chronic viral infection should be experimentally validated more vigorously.

In regards to Fig.1G, the claim of reduced Vb5+CXCR3+Neuropilin-1- iTregs in ifnar-/- mice is not convincing with only one flow plot shown, even if there is a modest reduction in the % of Vb5+ iTregs. Summary plots should be included. Moreover, given that the total # of Foxp3+ Treg cells are increased in ifnar-/- mice (shown in Sup. Fig.1G), the total number of Vb5+ iTregs is unlikely to differ. This should be discussed.

Do type I IFN levels (either mRNA or protein levels) correlate with the proportion or total number of

iTregs in LCMV-infected mice?

Can in vivo administration of either IFN β or poly IC (a potent agonizer of type I IFN production) further stimulate the induction of Vb5+ iTreg development during acute LCMV infection? Can this treatment potentially even induce Vb5+ iTreg development in MCMV or Vaccinia virus infection models, where they normally do not develop (sup. Fig1D)?

Did the authors consider performing adoptive transfer experiments using Ifnar^{-/-} VB5+ Foxp3⁻ T cells to ensure proper induction of type I IFN responses by other types of immune cells? Similarly, the authors could perform in vitro Treg skewing experiments using WT or Ifnar^{-/-} VB5+ Foxp3⁻ T cells with the addition of recombinant IFN β to determine the impact of CD4 intrinsic IFNAR signaling on iTreg development.

In my opinion, I don't think the tcrb^{-/-}tcrd^{-/-} model does justice to test the induction of autoimmunity. It seems like an artificial system. Preferably, I'd like the authors to use Foxp3DTR+IFNAR^{-/-} (1:1) +CD4^{-/-} to make mix BMC mice and deplete Tregs to see the impact on iTreg formation and determine if LCMV Cl13 can cause any autoimmune phenotypes. This is better than Figure1G given that IFNAR^{-/-} mice will be confounded by high viral load.

A major claim in this manuscript is that induction of Vb5+ iTregs is essential for protection against virus-induced colitis. However, although transfer of Vb5+ FoxP3+ CD4 T cells (which encompasses both iTregs and nTregs) can rescue the colitis pathology observed in chronic LCMV-infected tcrb^{-/-}tcrd^{-/-} mice, whether this is due to the functional activity of iTregs, nTregs, or both Treg subsets, remains unaddressed. The authors should have performed separate adoptive transfer experiments using either iTregs or nTregs to determine which subset is actually mediating this protection. Using CXCR3 and neuropilin-1 as markers to identify and sort these respective subsets would work. If transfer of iTregs is important for preventing colitis after LCMV infection, this would lend credence to their model.

Additionally, in the tcrb^{-/-}tcrd^{-/-} reconstitution model, is the protection afforded by CD4 T cells IFNAR-signaling dependent? Perhaps an anti-IFNAR blocking antibody could be used to test this?

Lastly, do Vb5-deficient mice develop colitis upon chronic LCMV infection? Can supplementation of Vb5+ iTregs prevent this? This would seem like a more adequate model as compared to using tcrb^{-/-}tcrd^{-/-} and reconstituting them with CD4 T cells.

I don't feel like the authors adequately addressed the mechanisms of how iTregs expand during infection. Instead of looking into IFN signaling, the paper turns toward antigen presentation and TCR strength. Both of which are very obvious to induce iTregs. Especially, when VB5+ Tregs are compromised, it seems there are some other lower affinity clones can compensate. Is there any physiopathological relevance in this?

Minor concerns

Figure legends for supplementary figure 1 were found incompatible with the figures and difficult to match.

Gating strategy for data in Figure 1B should be shown in Supplemental.

In regards to Figure 1, it would be helpful if the authors include flow data for chronic LCMV infection wherein VB5+Foxp3+ Tregs are first gated on, and then the relative distribution of CXCR3^{+/-}Neuropilin-1^{+/-} Tregs and iTregs are shown, so that we can better understand the relative proportions of iTregs to Tregs in this model.

In Figure 1D and S1B, to get TCRBV repertoire information, was it considered to perform TCR

repertoire sequencing on enriched TCR beta chains to confirm the repertoire information which has been retrieved here using Mixcr and contig assembly algorithm? Also, along with spleen it would be interesting to check the TCRBV repertoire in colon following LCMV infection.

In Figure 1F, should GFP be included with Foxp3 in axis label?

In terms of understanding suppressive activity of VB5+ iTreg subset, IL-10, CD25, CTLA4 (target of Foxp3) expression could have been checked with others in figure 2A and B.

In Figure 2F, there appears to be a significant increase in IFN- γ + CD8 T cells in the absence of Vb5+ Tregs, indicating that Tregs are likely suppressing the activity of virus-specific CD8 T cells to some degree, although viral titers remain similar. This should be briefly discussed rather than claiming that IFN- γ + CD8 T cells are comparable between control mice and mice that lacked Vb5+ CD4 T cells.

In figure 3C, statistics are missing between CD4-all vs Δ VB5 following LCMV infection in the reconstituted mice.

In figure 3D, did the authors consider checking other types of cytokines like TNF- α , IL-4, or IL-17 to check possible involvement of other types of CD4 effector T cell responses in response to colitogenic GC stimulation?

In figure 4A, 4B and S5B the information about scaling is missing.

In figure 4C, how to explain the splenomegaly in acute infection where CD4 and CD8 effector T cell responses were comparable, and induction of VB5+ iTreg was even higher in acute LCMV infection in mice where DCs were deficient of intracellular antigen processing compared to WT controls?

Reviewer #2, expert on intestinal T cell biology (Remarks to the Author):

The manuscript entitled “Rapid expansion of iTreg cells protects from collateral colitis following a viral trigger” proposes a fascinating and partially novel hypothesis: viral infections twist the immune tolerance from nTregs to iTregs. The latter cell type is fundamental to control a potentially pathological immune response in the intestine.

In particular, the major data presented in this manuscript show that upon LCMV infection, the number of Nrp1+ Foxp3 Treg cells reduces, while the number of the Nrp1- Foxp3 Treg cells increases over-time. In the total knock out mice for IFN α R, the Vb5+ iTreg cells do not expand.

In addition, Vb5+ T cells are not necessary to control LCMV infection, but they do cause pathological intestinal inflammation, which can be reversed by Vb5+ Foxp3+ Treg cells. Then the authors show the TCR repertoire of the CXCR3+ Foxp3+ Treg cells and using the ATG5 conditional knock out mice suggest that “exogenous antigens might be the primary driver” of Vb5+ Foxp3+ Treg cells.

Finally, they show that Vb5+ T cells have a higher TCR strength and they suggest that this might be the reason why they are so prone to acquire the expression of Foxp3. To improve the significance of their findings, the authors showed that there is an enrichment of TCRBV20-1 (B2) in human Foxp3 Treg cells isolated from healthy individuals.

In short, we believe that this paper has raised a very intriguing concept, but still some key aspects need to be improved.

Specific points:

Abstract: The abstract is well written and easy to follow.

Intro: Considering the data presented, we did not really understand why the intro focuses on RA and SLE and not on inflammatory bowel diseases (IBD).

We suggest rewriting the intro so that it focuses on IBD, in particular on (i) the role of Treg cells in IBD and, (ii) the connection among IFN α therapy, viral infection and IBD.

The summary of the results (from line 65 to 78) could be shortened in length and potentially expanded in the Discussion section.

Results:

Figure 1

A. **(Minor point)** At the beginning of the paragraph, the authors should explain what an acute infection is and what a chronic infection is. These concepts (acute and chronic) are only introduced in line 91 but they are already required to understand Fig.1A.

B. We suggest exploring (i.e. % and #) of Vb5 T effector and Treg cells in other tissues, such as the small and large intestines and Peyer's patches, considering that the manuscript will then explore the immune pathology in these tissues.

F. **(Minor point)** How the authors explain the low % of iTreg cells recovered.

G. This is a key point of the manuscript, but the data presented are weak at the moment. It would be much better to use a conditional Foxp3 or Lckd CRE mouse. If this is not possible, the authors should reconsider their conclusions accordingly. The concentration of IFNs over time in the blood should be checked.

- Are the iTregs spared by the depleting effect of IFNs because of time?
- In addition, how do the authors explain that only the Vb5 Treg cells are not expanding while the overall % iTreg is not altered? Which other clones expand instead?
- The figure 1SG is fundamental and it would be better to have it as a main figure.
- Figure 1SG: It would be important to explain why the # of Treg cells in WT mice does not oscillate over time as it does in figure 1a.

Figure 2

The authors observed a negative result and elegantly transformed it into a very interesting observation. This is the second key step of the manuscript. It would be important to show a positive control (e.g. a T cell clone able to reduce the f.f.u. upon transfer) in order to convince the readers that in principle, the experimental plan would allow them to see an effect on f.f.u. after T cell transfer.

Figure 3

The data presented here are very strong and interesting.

Figure 4

A – B. **(Minor point)** This part is not easy to follow in the text. For example: Trbv12-1-using CXCR3+ Treg cells also... (line 183-185). What does “using” mean? What is the key methodological difference between A and B. How was the clonal overlap calculated?

These important details are partially mentioned in the Methods, but it would be important to have some key explanations in the main text.

C. Comparing CRE+ vs CRE- is very risky, since the expression of CRE might have an effect per se. We suggest using CRE+ Atg5flox/flox vs littermates CRE+ Atg5 wt/wt. Was this mouse model previously validated?

What is the antigen repertoire that is presented by CD11c+ DCs in this mouse model?

How do the authors explain the splenomegaly?

Did they check the immune response in the intestine?

Did the mice develop intestinal inflammation?

(Minor point) The authors conclude that exogenous antigen might drive the expansion of these Tregs. Are these intestinal bacteria derived antigens? Treating the mice with antibiotics could be a way to test this. Using germ-free mice would be ideal.

Figure 5

(Minor point) It would be interesting to explore which kind of Vb5+ T cells are able to convert into Treg cells? Naïve, effector (e.g. Th1, Th17, Tfh), or memory cells?

I-J. These data remain very superficial. We would suggest, if possible, checking in IBD patients the relative expression of Vb2 cells in Treg cells. Alternatively, you could check in viral infected patients, but ideally, checking in both (IBD and infection) would be the best.

(Minor point): It would be important to explicitly write what “var” stands for.

Discussion.

As with the intro, it would be more appropriate to discuss the data in the context of IBD instead of SLE or RA. Alternative experimental data investigating the link between LCMV and SLE or RA should be presented.

IBD is not a classical autoimmune disease and this should also be discussed considering that the bacterial derived antigens are most probably the relevant antigens.

Point-by-Point Reply

Manuscript NCOMMS-19-09838-T: Rapid expansion of Treg cells protects from collateral colitis following a viral trigger

We would like to thank the reviewers for their constructive criticism. We have been able to address the reviewers' concerns with addition of new experimental data, which has further strengthened the manuscript. The detailed description on how we have addressed the issues raised is outlined in the point-by-point response below.

For a better overview, we would like to quickly summarize the main points and conceptual advance of the paper upfront:

- The anti-viral type I IFN response was previously shown to transiently reduce Tregs numbers. We show here that this loss is rapidly compensated by Tregs carrying the V β 5 TCR.
- This V β 5⁺ Treg population is essential for maintaining self-tolerance after viral infection, as their depletion delays the replenishment of the Treg niche, resulting in colitis.
- Colitis can be prevented by adoptive transfer of V β 5⁺ Tregs but not other Treg subsets.
- The unique ability of V β 5⁺ T cells to rapidly replenish the Treg niche is anchored in an intrinsically higher TCR signal than in other T cells, increasing the ability of V β 5⁺ T cells to proliferate and expand as well as to convert into iTregs.
- V β 2⁺ T cells represent the corresponding T cell population in humans and are expanded in patients recovering from active colitis.

These results therefore provide a missing link between an infectious trigger and the initiation of a pathogenic inflammatory response (colitis) that does not share antigen with the infectious agent and thus provide an explanation for how viral infections may serve as an environmental trigger for inflammatory disorders and autoimmunity.

Reviewer #1, expert on LCMV and T cells

Major concerns:

Given that a major premise of this work is based on viral-induced type I IFN-mediated induction of iTregs, and their subsequent role in preventing autoimmunity, this assertion, and the biological relevance iTreg expansion during chronic viral infection should be experimentally validated more vigorously.

In regards to Fig.1G, the claim of reduced Vb5+CXCR3+Neuropilin-1- iTregs in *ifnar*^{-/-} mice is not convincing with only one flow plot shown, even if there is a modest reduction in the % of Vb5+ iTregs. Summary plots should be included. Moreover, given that the total # of Foxp3+ Treg cells are increased in *ifnar*^{-/-} mice (shown in Sup. Fig.1G), the total number of Vb5+ iTregs is unlikely to differ. This should be discussed.

As suggested by the reviewer, we have replaced the flow plot by summary graphs for the % and total number of Vβ5⁺ Tregs and found a strong reduction of Vβ5⁺ Tregs in *Ifnar1*^{-/-} mice as well as in WT CD4⁺ T cells that were transferred into *Ifnar1*^{-/-} mice (new Figure 1H of the revised manuscript). This data confirm a strong reduction of Vβ5⁺ Tregs in *Ifnar1*^{-/-} mice and further demonstrate that this is not due to direct effects of type I IFN on Vβ5⁺ Tregs. It is rather the fact that *Ifnar1*^{-/-} mice do not show a reduction of Tregs upon LCMV infection and thus Vβ5⁺ Tregs, which have the unique ability to rapidly replenish the Treg compartment to prevent pathology, do not expand.

Do type I IFN levels (either mRNA or protein levels) correlate with the proportion or total number of iTregs in LCMV-infected mice?

Can *in vivo* administration of either IFN β or poly IC (a potent agonizer of type I IFN production) further stimulate the induction of Vb5+ iTreg development during acute LCMV infection? Can this treatment potentially even induce Vb5+ iTreg development in MCMV or Vaccinia virus infection models, where they normally do not develop (sup. Fig1D)?

The reviewer raised an interesting point and we have performed acute LCMV and Vaccinia Virus infections with and without polyIC administration to augment the type I IFN response (chronic LCMV was included as a positive control). We have measured the type I IFN levels in each group and correlated this with the proportion of Vβ5⁺ Tregs. While the type I IFN levels that can be induced *in vivo* using polyIC are far lower than what is observed upon chronic LCMV infection, we nonetheless observed a small increase (see Figure I below). Although this increase did not induce Vβ5⁺ Treg frequencies comparable to those observed in chronic LCMV infection, we could observe a positive correlation of type I IFN levels with Vβ5⁺ Treg frequencies (Figure 1G of the revised manuscript).

Figure I: Type I levels over the course of viral infection. C57BL/6 WT mice were infected with LCMV WE (200 f.f.u. i.v.), Clone 13 (2x10⁶ FFU i.v.) or Vaccinia virus (2x10⁶ FFU i.v.). In addition, indicated groups received 50μg polyIC on days 0 and 2 after infection. Left: Type I interferon levels were determined from serum by luciferase-based bioassay over the course of infection, (n=5). Right: type I IFN levels at 24h were correlated with Vβ5⁺ Treg frequencies on day 10.

Did the authors consider performing adoptive transfer experiments using *Ifnar*^{-/-} Vβ5⁺ Foxp3⁻ T cells to ensure proper induction of type I IFN responses by other types of immune cells?

We thank the reviewer for this interesting suggestion. We do not have the *Ifnar*^{-/-} mice crossed to the Foxp3-GFP.KI reporter mice and as such the isolation of a pure effector T cell population is difficult. Nevertheless, we have generated mixed bone marrow (bm) chimeras to address this point: WT recipients were reconstituted with *Ifnar*^{-/-} bm + CD4^{-/-} bm (1:1) so that in the resulting mice all T cells are *Ifnar*^{-/-}, while for all other cell types at least half the cells can sense type I IFN, ensuring a proper type I IFN response. Unfortunately, we had considerable technical difficulties with this approach as the LCMV infection was not tolerated well in the chimeras and a sizeable proportion of mice had to be euthanized before they could be analyzed. In addition, the results obtained from the (few) remaining mice in the three repeats we performed were quite variable and we do not feel confident including the results in the manuscript. Nevertheless, the results suggest that the *Ifnar*^{-/-} chimeras harbor reduced frequencies of iTregs as the full *Ifnar*^{-/-} mice do (included below for your reference as Figure II). Furthermore, as outlined in the next point, we have shown that the type I IFN does not directly act on the T cells but rather has an indirect effect by depleting the Treg niche (Fig. 1H and S2G of the revised manuscript).

Figure II: iTreg induction in mixed bone marrow chimeras. WT recipients were reconstituted 6-8 weeks with *Ifnar*^{-/-}+CD4^{-/-} (1:1) or WT+CD4^{-/-} (1:1) bm and chronically infected with 10⁶ f.f.u. LCMV clone 13 and iTreg frequencies were determined by flow cytometry on day 10 post infection (pooled data from 3 experiments).

Similarly, the authors could perform *in vitro* Treg skewing experiments using WT or *Ifnar*^{-/-} Vβ5⁺ Foxp3⁻ T cells with the addition of recombinant IFNβ to determine the impact of CD4 intrinsic IFNAR signaling on iTreg development.

We thank the reviewer for this valuable suggestion. We have performed classical *in vitro* iTreg differentiations with and without type I IFN but could not observe an effect on the conversion of Vβ5⁺ (or other) T cells into iTregs (Figure S2G of the revised manuscript). In addition, we have transferred congenically marked WT CD4⁺ T cells into *Ifnar*^{-/-} mice before LCMV infection. In this setting, type I IFN is produced but cannot be sensed by endogenous T cells, preventing Treg loss. However, the type I IFN response elicited upon LCMV infection did not result in the expansion of Vβ5⁺ Tregs, demonstrating that type I IFN also does not directly promote Vβ5⁺ iTreg conversion or expansion *in vivo*. Finally, we have reconstituted *Tcrb*^{-/-}*Tcrd*^{-/-} mice with Vβ5⁻ WT CD4⁺ T cells + Vβ5⁺CD4⁺ *Ifnar*^{-/-} T cells. In this setting, the Vβ5⁺ T cells cannot sense type I IFN but are still able to expand and protect from colitis (revised Figure 3A). These experiments conclusively demonstrate that type I IFN do not directly promote Vβ5⁺ iTreg conversion *in vitro* or *in vivo*.

In my opinion, I don't think the *tcrb*^{-/-}*tcrd*^{-/-} model does justice to test the induction of autoimmunity. It seems like an artificial system. Preferably, I'd like the authors to use Foxp3DTR+IFNAR^{-/-} (1:1) +CD4^{-/-} to make mix BMC mice and deplete Tregs to see the impact on iTreg formation and determine if LCMV Cl13 can cause any autoimmune phenotypes. This is better than Figure1G given that IFNAR^{-/-} mice will be confounded by high viral load.

We agree with the reviewer that the T cell transfer model has some limitations and Vβ5-deficient

mice as suggested by the reviewer further below would be an ideal model to address the function of $V\beta 5^+$ expressing $CD4^+$ T cells and Tregs. However, as outlined in detail below, we do not have access to these mice nor are we aware of an existing live mouse strain that is $V\beta 5^-$ deficient. In light of this, we think that our reconstitution approach in T cell-deficient mice serves as a good model to functionally address the relevance of $V\beta 5^+$ expressing $CD4^+$ T cells in LCMV infection. Importantly, as our data shows the model works nicely and the uninfected mice show no phenotype (Figure 3).

Nevertheless, we have addressed the point raised by the reviewer from several angles. On one hand, we have shown that type I IFN does not directly induce conversion of $V\beta 5^+$ $CD4^+$ T cells into Tregs both *in vitro* (Figure S2G) and *in vivo* (Figure 1H and 3A). On the other hand, we agree with the reviewer that a bone marrow chimera approach is more elegant than the full *Ifnar1^{-/-}* and have generated two sets of bone marrow chimeras:

1. *Ifnar1^{-/-}* bm + DREG bm (1:1) into WT recipients, with continuous DT treatment throughout the infection; in these mice all Tregs are *Ifnar1^{-/-}*, while for all other cell types at least half the cells can sense type I IFN, ensuring a proper induction of the immune response. Unfortunately, this setup generated considerable technical issues as the bm chimeras didn't tolerate the continuous administration of DT in combination with the LCMV infection and had to be euthanized before results could be obtained. As such, this approach could not be pursued further.
2. *Ifnar1^{-/-}* bm + $CD4^+$ bm (1:1) into WT recipients; in these mice all $CD4^+$ T cells are *Ifnar1^{-/-}*, while for all other cell types at least half the cells can sense type I IFN. While the effect is not limited to Tregs here it is still much more defined than in full *Ifnar1^{-/-}* mice. As outlined above, we obtained variable results so that we did not include them in the manuscript, but they are shown in Figure II above and the trend observed is in line with the model we propose.

Finally, we have selectively removed the type I IFN receptor from $V\beta 5^+$ T cells by reconstituting *Tcrb^{-/-}Tcrd^{-/-}* mice with $V\beta 5^+$ WT $CD4^+$ T cells + $V\beta 5^+$ $CD4^+$ *Ifnar1^{-/-}* T cells. In this setting, the $V\beta 5^+$ T cells cannot sense type I IFN but are still able to expand and protect from colitis (revised Figure 3A). These experiments confirm that the type I IFN acts on the T cells (to reduce Treg numbers) but does not directly promote conversion of $V\beta 5^+$ T cells into iTregs but rather depleted the Treg niche, which allows for their preferential conversion and expansion.

*A major claim in this manuscript is that induction of $V\beta 5^+$ iTregs is essential for protection against virus-induced colitis. However, although transfer of $V\beta 5^+$ FoxP3⁺ $CD4^+$ T cells (which encompasses both iTregs and nTregs) can rescue the colitis pathology observed in chronic LCMV-infected *tcrb^{-/-}tcrd^{-/-}* mice, whether this is due to the functional activity of iTregs, nTregs, or both Treg subsets, remains unaddressed. The authors should have performed separate adoptive transfer experiments using either iTregs or nTregs to determine which subset is actually mediating this protection. Using CXCR3 and neuropilin-1 as markers to identify and sort these respective subsets would work. If transfer of iTregs is important for preventing colitis after LCMV infection, this would lend credence to their model.*

We agree with the reviewer that it would be interesting to address the relative ability of $V\beta 5^+$ iTregs vs. $V\beta 5^+$ nTregs to rescue mice from the colitis pathology. However, given the reduced numbers of Tregs recovered from an LCMV infected mouse, 10-15 mice are required to isolate approximately 100'000 $V\beta 5^+$ nTregs required for a single recipient. Based on our rough estimation this will translate into 50-60 mice and 20 hours of sort time for a single repeat of the experiment. In addition, we have observed that binding of the anti-neuropilin-1 antibody is not stable and the staining fluctuates on unfixed cells and thus does not work reliably for sorting. On this basis, we think it is technically impossible to compare $V\beta 5^+$ iTregs and $V\beta 5^+$ nTregs in this experimental approach. In addition, we would like to emphasize that we do not claim that $V\beta 5^+$

nTregs are not capable of preventing colitis, but this (and other) Treg populations are simply not present in sufficient numbers after the LCMV challenge to control the colitogenic immune response. We realize that this might have been confusing and revised the text and title to improve this.

Despite the technical limitations, we have tried to approach the reviewer's question from several angles to strengthen our findings:

1. We have reconstituted *Tcrb*^{-/-}*Tcrd*^{-/-} mice with Vβ5⁻ WT CD4⁺ T cells + Vβ5⁺ DREG CD4⁺ T cells and treated with DT upon infection; these mice have no Vβ5⁺ Tregs but harbor Vβ5⁺ effector CD4⁺ T cells and thus the Vβ5 defect is limited to the Treg compartment. Just like in the ΔVβ5 group, these mice develop colitis, because the Vβ5⁺ Tregs are depleted upon infection, which is in line with an essential role of Vβ5⁺ Tregs in replenishing the Treg niche to prevent colitis (revised Figure 3A).
2. We have adoptively transferred Vβ8⁺ Tregs as an alternative Treg subset to try to rescue the colitis in ΔVβ5 reconstituted *Tcrb*^{-/-}*Tcrd*^{-/-} mice. Interestingly, the Vβ8⁺ Tregs were not able to protect from colitis like the Vβ5⁺ Tregs. We also recovered slightly lower numbers of Tregs from these mice indicating that the Vβ8⁺ Tregs cannot expand as vigorously as the Vβ5⁺ Tregs and are thus less potent in this setting (revised Figure 3A).

Together, these additional experiments confirm the essential role of Vβ5⁺ Tregs in preventing colitis upon LCMV infection.

Additionally, in the tcrb^{-/-}tcrd^{-/-} reconstitution model, is the protection afforded by CD4 T cells IFNAR-signaling dependent? Perhaps an anti-IFNAR blocking antibody could be used to test this?

As mentioned above, our additional data confirms that type I IFN does not directly induce conversion of Vβ5⁺ CD4⁺ T cells into Tregs both *in vitro* (Figure S2G) or *in vivo* (Figure 1H). Furthermore, we do not observe colitis in the condition where CD4⁺ T cells afford protection in the *Tcrb*^{-/-}*Tcrd*^{-/-} model (CD4 all group, Figure 3A). In the absence of type I IFN signaling there is no loss of Tregs (as in the *Ifnar1*^{-/-} mice), so again, there is no colitis. If anything, blocking type I IFN would be even more protective, but we already see full protection without. Nevertheless, we have reconstituted *Tcrb*^{-/-}*Tcrd*^{-/-} mice with Vβ5⁻CD4⁺ WT T cells + Vβ5⁺CD4⁺ *Ifnar1*^{-/-} T cells and found that these mice were protected from colitis (revised Figure 3A, B), demonstrating that direct IFNAR signaling into Vβ5⁺CD4⁺ T cells is not required for protection.

Lastly, do Vb5-deficient mice develop colitis upon chronic LCMV infection? Can supplementation of Vb5+ iTregs prevent this? This would seem like a more adequate model as compared to using tcrb^{-/-}tcrd^{-/-} and reconstituting them with CD4 T cells.

We agree with the reviewer that using Vβ5-deficient mice would be a straightforward approach to elucidate the importance of Vβ5⁺ T cells in preventing immune pathology in response to LCMV infection. However, we do not have access to these mice nor are we aware of an existing live mouse strain. In light of this, we think that our reconstitution approach in T cell deficient mice nicely circumvents this problem and - as our data shows - serves as a good model to functionally address the relevance of Vβ5⁺ expressing CD4⁺ T cells in LCMV infection.

I don't feel like the authors adequately addressed the mechanisms of how iTregs expand during infection. Instead of looking into IFN signaling, the paper turns toward antigen presentation and TCR strength. Both of which are very obvious to induce iTregs. Especially, when VB5+ Tregs are compromised, it seems there are some other lower affinity clones can compensate. Is there any physiopathological relevance in this?

As our additional data shows type I IFN does not directly promote the conversion of V β 5⁺ T cells into Tregs (Figure 1H and S2G). However, we did investigate the factors that drive the replenishment of the Treg pool. As outlined in Figure 5, V β 5⁺CD4⁺ T cells show higher TCR signaling (Fig. 5D, E), higher conversion rates into Tregs (Fig. 5F), as well as higher proliferation rates (Fig. 5C). All of these aspects contribute to an accelerated replenishment of the Treg pool (Fig. 5A, day 10). While other clones can eventually compensate (Fig. 5A, day 17), we clearly observe a prolonged decrease in Tregs in the absence of V β 5⁺CD4⁺ T cells that is accompanied by colitis (Fig. 3A). The fact that adoptive transfer of V β 5⁺ Tregs during this period can reverse colitis (Fig. 3A) further shows that the loss of Tregs drives the pathology. Furthermore, our newly generated data shows that V β 8⁺ Tregs are not capable a fully rescuing the colitis phenotype (Fig. 3A). As such, the physiological relevance is clear and other clones cannot compensate as rapidly as the V β 5⁺ clones do. In addition to the data we obtained in the mouse model, we have now also investigated the V β usage in colitis patients with active and inactive disease and compared these with healthy controls (new Figure 5J-K of the revised manuscript). As V β 5⁺ T cells in mice, V β 2⁺ T cells in humans are overrepresented in the Treg compartment (Figure 5I). In line with the predominant expansion of V β 5⁺ Tregs in mice we observed a higher frequency of V β 2⁺ Tregs in colitis patients with inactive disease (new Figure 5J,K). This is also in line with previous reports that have observed higher frequencies of V β 2⁺ Tregs in colitis patients (Zeissig et al, *Gut*, 2019). Overall, our data clearly support a physiological role of a specific T cell subset in replenishing a compromised Treg niche and in preventing colitis.

Minor concerns

Figure legends for supplementary figure 1 were found incompatible with the figures and difficult to match.

We apologize for this error and have corrected this in the revised manuscript.

Gating strategy for data in Figure 1B should be shown in Supplemental.

The gating strategy has been included in the revised manuscript (Fig. S1C and Figure III below).

In regards to Figure 1, it would be helpful if the authors include flow data for chronic LCMV infection wherein VB5+Foxp3+ Tregs are first gated on, and then the relative distribution of CXCR3+/- Neuropilin-1+/- Tregs and iTregs are shown, so that we can better understand the relative proportions of iTregs to Tregs in this model.

We have performed the analysis in both ways, i.e. first gating on iTregs then analyzing the TCR V β usage or first gating on T cell subsets carrying a specific TCR V β chain and then analyzing iTreg vs. nTreg contribution, and obtained the same results with both approaches (Figure III below). There is a relative increase of V β 5⁺ iTregs upon LCMV infection.

Figure III. Gating strategies for Vβ5⁺ iTregs. WT mice were infected with 10⁶ f.f.u. LCMV clone 13 and Treg cells were analyzed on day 14 by flow cytometry. Gating strategy 1 was used throughout the paper (top), gating strategy 2 was suggested by the reviewer (bottom).

In Figure 1D and S1B, to get TCRBV repertoire information, was it considered to perform TCR repertoire sequencing on enriched TCR beta chains to confirm the repertoire information which has been retrieved here using Mixcr and contig assembly algorithm? Also, along with spleen it would be interesting to check the TCRBV repertoire in colon following LCMV infection.

We did not perform repertoire sequencing on enriched populations. However, according to the reviewer's suggestion, we have evaluated the proportion of Vβ5⁺CD4⁺ T cells in comparison to other subsets in the colon and the mesenteric LN and include this information in the additional Figure S2F of the revised manuscript. We detected an increase in Vβ5⁺ iTreg frequencies in the colon that is consistent with the systemic increase observed. However, cell numbers were very variable so total numbers should be interpreted with caution.

In Figure 1F, should GFP be included with Foxp3 in axis label?

We thank the reviewer for this helpful comment but in the sample shown, the analysis was performed by conventional staining with anti-Foxp3 antibody.

In terms of understanding suppressive activity of Vβ5⁺ iTreg subset, IL-10, CD25, CTLA4 (target of Foxp3) expression could have been checked with others in figure 2A and B.

We agree with the reviewer, that these are important mediators of suppression and have included the analysis of CTLA-4 and IL-10 in the revised manuscript (Figure S2D-E).

In Figure 2F, there appears to be a significant increase in IFN-γ⁺ CD8 T cells in the absence of Vβ5⁺ iTregs, indicating that Tregs are likely suppressing the activity of virus-specific CD8 T cells to some degree, although viral titers remain similar. This should be briefly discussed rather than claiming that IFN-γ⁺ CD8 T cells are comparable between control mice and mice that lacked Vβ5⁺ CD4 T cells.

We have revised the results section to point out the small difference in CD8⁺ T cells. We have

also performed additional experiments to determine the role of CD8⁺ T cells in the colitogenic process and found that they are essential drivers of the inflammation as *Tcrb*^{-/-}*Tcrd*^{-/-} mice that do not receive CD8⁺ T cells are protected from colitis. This important finding is included in the revised Figure 3A of the manuscript.

In figure 3C, statistics are missing between CD4-all vs ΔVB5 following LCMV infection in the reconstituted mice.

We have revised Figure 3C to include the statistics.

In figure 3D, did the authors consider checking other types of cytokines like TNF-α, IL-4, or IL-17 to check possible involvement of other types of CD4 effector T cell responses in response to colitogenic GC stimulation?

We have also tested for TNF-α, IL-4, and IL-17 in this experiment and now also included this data in the revised manuscript as supplementary Figure 4.

In figure 4A, 4B and S5B the information about scaling is missing.

We thank the reviewer for pointing this out and have revised the figure legends accordingly.

In figure 4C, how to explain the splenomegaly in acute infection where CD4 and CD8 effector T cell responses were comparable, and induction of VB5+ iTreg was even higher in acute LCMV infection in mice where DCs were deficient of intracellular antigen processing compared to WT controls?

These mice have been characterized previously and show strongly increased antigen presentation, resulting in enhanced cell expansion, which in the case of LCMV infection translates into splenomegaly (Loi M. et al, *Cell Reports*, 2016).

Reviewer #2, expert on intestinal T cell biology

Specific points: □

Abstract: *The abstract is well written and easy to follow.*

Intro: *Considering the data presented, we did not really understand why the intro focuses on RA and SLE and not on inflammatory bowel diseases (IBD). □ We suggest rewriting the intro so that it focuses on IBD, in particular on (i) the role of Treg cells in IBD and, (ii) the connection among IFNα therapy, viral infection and IBD.*

The summary of the results (from line 65 to 78) could be shortened in length and potentially expanded in the Discussion section.

We thank the reviewers for this suggestion and have edited the introduction and discussion to include specific reference to IBD.

Results:

Figure 1

A. (Minor point) *At the beginning of the paragraph, the authors should explain what an acute infection is and what a chronic infection is. These concepts (acute and chronic) are only introduced in line 91 but they are already required to understand Fig. 1A.*

We thank the reviewers for pointing this out. We have revised the paragraph to introduce LCMV in detail already at this point of the manuscript.

B. *We suggest exploring (i.e. % and #) of Vb5 T effector and Treg cells in other tissues, such as the small and large intestines and Peyer's patches, considering that the manuscript will then*

explore the immune pathology in these tissues.

We have extended our analysis of V β 5⁺ Treg subsets to the mesenteric LN and the colon (Figure S2F of the revised manuscript) and found V β 5⁺ iTreg frequencies in the colon to be increased with LCMV infection, which is consistent with our systemic observations.

F. (Minor point) How the authors explain the low % of iTreg cells recovered.

Lymphoreplete hosts show a very low rate of iTreg induction, which is thought to be at least partially due to the insufficient amounts of antigens that drive Treg cell selection. The frequencies we observed are comparable to those observed by others when not transferring into lymphopenic mice (e.g. Hand TW, *Science*, 2012).

G. This is a key point of the manuscript, but the data presented are weak at the moment. It would be much better to use a conditional Foxp3 or Lckd CRE mouse. If this is not possible, the authors should reconsider their conclusions accordingly.

Unfortunately, we do not have access to these mice but have addressed this important point from several angles to strengthen it. We generated mixed bone marrow (bm) chimeras where WT recipients were reconstituted with *Ifnar1*^{-/-} (or WT ctrl) bm + CD4^{-/-} bm (1:1); in these mice all CD4⁺ T cells are *Ifnar1*^{-/-}, while for all other cell types at least half the cells can sense type I IFN. Unfortunately, we had considerable technical difficulties with this approach as the LCMV infection was not tolerated well in the chimeras and a sizeable proportion of mice had to be euthanized before they could be analyzed. In addition, the results obtained from the remaining mice in the three repeats we performed were quite variable and we do not feel confident including the results in the manuscript. Nevertheless, the results suggest that the *Ifnar1*^{-/-} chimeras harbor reduced frequencies of iTregs as the full *Ifnar1*^{-/-} mice do and we have included them for your reference as Figure II (see response to reviewer 1 above). Furthermore, we have shown that the type I IFN does not directly act on the T cells but rather has an indirect effect by depleting the Treg niche. To this end we have performed classical *in vitro* iTreg differentiations with and without type I IFN but could not observe an effect on the conversion of V β 5⁺ (or other) T cells into iTregs (Figure S2G of the revised manuscript). In addition, we have transferred congenically marked WT CD4⁺ T cells into *Ifnar1*^{-/-} mice before LCMV infection. In this setting, type I IFN is produced but cannot be sensed by endogenous T cells, preventing Treg loss. However, the type I IFN response elicited upon LCMV infection did not result in the expansion of V β 5⁺ Tregs, demonstrating that type I IFN also does not directly promote V β 5⁺ iTreg conversion or expansion *in vivo*. Finally, we have correlated the type IFN levels after different viral infections with the magnitude of the V β 5⁺ iTreg response and see a highly significant positive correlation (new Figure 1G). Taken together, these new experiments substantially strengthen our point and show that a strong type I IFN response is necessary to induce the expansion of V β 5⁺ iTregs but that the effect is not direct, rather type I IFN compromises the Treg compartment which is then replenished with V β 5⁺ Tregs.

The concentration of IFNs over time in the blood should be checked. □- Are the iTregs spared by the depleting effect of IFNs because of time? □-

We agree with the reviewers that it is important to look at type I IFN levels over time and have included this as Supplementary Figure S2D of the revised manuscript. After an initial peak at 24 hours, lower levels of type I IFN are maintained in chronically infected mice over the period analyzed. Hence we find a continuous presence of type I IFN and as such it seem unlikely that iTregs are spared because of differential kinetics.

In addition, how do the authors explain that only the Vb5 Treg cells are not expanding while the overall % iTreg is not altered? Which other clones expand instead? □-

First of all, we do see a certain expansion of all Tregs, including $V\beta 5^+$ Tregs. However, as there is no type-I IFN-dependent loss in Tregs in this setting, we observe a generalized activation and expansion of all Tregs without a preferential $V\beta 5^+$ Treg expansion.

Most infectious challenges lead to an increase in iTregs in response to the infection. This iTreg induction is driven through TCR stimulation in combination with polarizing factors such as TGF- β or retinoic acid. As such the iTreg pool is generally largely composed of pathogen-specific iTregs. In the case of LCMV infection in WT mice, the Treg compartment is strongly depleted due to type I IFN generated in response to the viral infection (Srivastava et al., *J Exp Med*, 2014; Gangaplara et al., *PLoS Pathog*, 2018). Here, the Treg niche is replenished mostly with $V\beta 5^+$ iTregs and the replenishment is partially driven through lymphopenia-induced proliferation and activation. This does not occur in *Ifnar1^{-/-}* mice, as there is no loss of Tregs (Figure 1H and Supplementary Figure 2F). Hence there is no over-proportional expansion of the $V\beta 5^+$ iTregs and the iTreg compartment in *Ifnar1^{-/-}* mice and their Treg compartment is likely largely composed of pathogen-specific iTregs.

The figure 1SG is fundamental and it would be better to have it as a main figure. - Figure 1SG: It would be important to explain why the # of Treg cells in WT mice does not oscillate over time as it does in figure 1a.

We agree that this is an important point and as outlined above we have substantially revised this part of the manuscript to strengthen the connection between type I IFN and $V\beta 5^+$ Treg accumulation. As such we have strongly reshaped Figure 1 and S1-S2 and feel that the newly generated data in Fig. 1G and 1H even are most important and informative. Also, as the type I IFN-dependent Treg loss was already reported in two previous publications (Srivastava et al., *J Exp Med*, 2014; Gangaplara et al., *PLoS Pathog*, 2018), it is of limited novelty and we have focused on the most important new findings in the main figures.

The kinetic shown in the old Figure S1G (Figure S2F of the revised manuscript) is not as tight as that in figure 1A. The earliest time point shown in figure S1G is day 10, by which Treg numbers have already largely recovered (Fig. 1A).

Figure 2

The authors observed a negative result and elegantly transformed it into a very interesting observation. This is the second key step of the manuscript. It would be important to show a positive control (e.g. a T cell clone able to reduce the f.f.u. upon transfer) in order to convince the readers that in principle, the experimental plan would allow them to see an effect on f.f.u. after T cell transfer.

We have determined the viral loads in LCMV-infected DERE mice, in which Tregs were depleted and observed complete viral clearance in the absence of Tregs due to an augmented effector response (new Figure 2G of the revised manuscript). This is also in line with the literature, where Tregs were shown to be important to promote viral persistence (Schmitz, I et al., *PLoS Pathog*, 2013).

Figure 3

The data presented here are very strong and interesting.

We thank the reviewers for this positive feedback.

Figure 4 □A – B. (Minor point) This part is not easy to follow in the text. For example: *Trbv12-1-using CXCR3+ Treg cells also...* (line 183-185). What does “using” mean? What is the key methodological difference between A and B. How was the clonal overlap calculated? □ These important details are partially mentioned in the Methods, but it would be important to have some key explanations in the main text.

We thank the reviewers for pointing this out and have revised the text to improve readability.

C. Comparing CRE+ vs CRE- is very risky, since the expression of CRE might have an effect per se. We suggest using CRE+ Atg5flox/flox vs littermates CRE+ Atg5 wt/wt. Was this mouse model previously validated? What is the antigen repertoire that is presented by CD11C+ DCs in this mouse model? How do the authors explain the splenomegaly?

The reviewers have raised an important point and we have repeated the experiments with both Cre⁻Atg5^{fl/fl} as well as Cre⁺Atg5^{fl/wt} mice as controls and have observed no differences between Cre⁺Atg5^{fl/wt} and Cre⁺Atg5^{wt/wt} mice (see figure III below). The mouse model has already been described (Loi, M. et al., *Cell Reports*, 2016) and we have added this reference in the revised manuscript. In these mice, antigen presentation is strongly increased, resulting in augmented responses, which in the case of LCMV infection translates into splenomegaly.

Figure III. Vβ5⁺iTreg induction in Cre⁺Atg5^{fl/wt} and Cre⁺Atg5^{wt/wt} mice. Mice were infected with LCMV Clone 13 (2x10⁶ f.f.u i.v.). Half the animals continuously received antibiotics in the drinking water starting from 2 weeks before infection. On day 10 p.i. the frequency of Vβ5⁺ iTregs in the spleen and mLN was quantified by flow cytometry, (n=2-5).

The antigen repertoire in Atg5^{fl/fl}xCD11c-Cre mice has not been fully characterized but we are aware of other groups that are currently exploring this question under homeostatic conditions. In the context of LCMV infection, we do not think that the expansion of Vβ5⁺ Tregs can be attributed to one specific antigen but rather to a combination of factors leading to the observed phenotype (compromised Treg niche and thus lymphopenia-induced proliferation, enhanced TCR signaling in Vβ5⁺ T cells, enhanced conversion of Vβ5⁺ T cells into iTregs). Nevertheless, we have performed an analysis of the antigenic peptide repertoire using a proteomics approach.

[Redacted]

In summary, our analysis of the MHC Class II peptidome was not conclusive and we could not identify candidate antigens, which might drive the expansion of Vβ5⁺ Tregs. However, as pointed out above, **we do not think that the expansion of Vβ5⁺ Tregs can be attributed to one specific antigen** but is the result of multiple factors that favor conversion of Vβ5⁺ T cells into Tregs and their subsequent expansion.

[Redacted]

Did they check the immune response in the intestine? Did the mice develop intestinal inflammation?

Given that mice lacking Vβ5⁺CD4⁺ T cells develop colitis, we did indeed look for indications of intestinal inflammation in these mice but did not find any (see Figure V below). We also did not

observe any difference in frequencies of $V\beta 5^+CD4^+$ Tregs or in cytokine secretion by effector T cells in the mesenteric LN. Given that these mice do not show a phenotype for intestinal inflammation, we did not include this information in the manuscript.

Figure V: Mice with autophagy-deficient DCs do not show intestinal pathology. $Cre^+Atg5^{fl/fl}$, $Cre^+Atg5^{fl/wt}$ and $Cre^-Atg5^{fl/fl}$ mice were infected LCMV Clone 13 (2×10^6 f.u.). Half the animals received continuous antibiotics treatment via the drinking water starting two weeks before infection. On day 10 post infection the histopathological changes in the A) ceacum and B) colon were evaluated by H&E staining (n=3-5/group).

(Minor point) The authors conclude that exogenous antigen might drive the expansion of these Tregs. Are these intestinal bacteria derived antigens? Treating the mice with antibiotics could be a way to test this. Using germ-free mice would be ideal.

According to the reviewers suggestion, we have treated the mice with antibiotics but did not see a difference in the induction of $V\beta 5^+$ Tregs, suggesting that unlike the colitogenic $CD8^+$ T cell response, induction of $V\beta 5^+$ iTreg cells is not driven by microbial antigens. We have included this data as Supplementary Figure S5C-E of the revised manuscript.

Figure 5 *(Minor point) It would be interesting to explore which kind of $V\beta 5^+$ T cells are able to convert into Treg cells? Naïve, effector (e.g. $Th1$, $Th17$, Tfh), or memory cells?*

Based on the literature and our own experience, memory T cells or differentiated effector T cells are not able to differentiate into iTregs. This is illustrated in Figure VI below. As such we think that conversion of naïve $V\beta 5^+CD4^+$ T cells into iTregs drives the expansion of the $V\beta 5^+$ Treg population.

Figure VI: Memory T cells do not convert into iTregs. Naïve ($CD44^+CD62L^+$) and memory ($CD44^+CD62L^-$) $CD4^+$ T cells were sorted from memory mice and stimulated with plate bound anti- $CD3/CD28$ under iTreg polarizing conditions (3ng/ml $TGF-\beta$). Foxp3 induction was determined by flow cytometry on day 5.

I-J. These data remain very superficial. We would suggest, if possible, checking in IBD patients the relative expression of $V\beta 2$ cells in Treg cells. Alternatively, you could check in viral infected patients, but ideally, checking in both (IBD and infection) would be the best.

We agree with the reviewers that patient data would strongly support the concept presented and have analyzed the effector T cell and Treg compartment of IBD patients (active and inactive disease) and compared it to healthy controls. It was not possible to extend our analysis to

patients undergoing a viral infection, as we were not be able to obtain samples from before the infection to compare the frequencies to. We thus focused our analysis on IBD patients and analyzed V β 2 expression in Tregs vs. effector T cells by FACS. In line with the relative overrepresentation of V β 2⁺ T cells in the Treg compartment (Fig. 5I), we found elevated frequencies of V β 2⁺ Tregs in IBD patients with inactive disease, indicative of a recent Treg expansion, which would dampen active disease (Figure 5J-K of the revised manuscript). This data fully supports our reported finding that a dedicated T cell subset in both mice and humans displays a superior ability to replenish the Treg pool if it has been compromised.

(Minor point): It would be important to explicitly write what “var” stands for.

This is defined in the gating strategy (Fig. S5) but we have changed the axis labels to make this less confusing.

Discussion.

As with the intro, it would be more appropriate to discuss the data in the context of IBD instead of SLE or RA. Alternative experimental data investigating the link between LCMV and SLE or RA should be presented. □ IBD is not a classical autoimmune disease and this should also be discussed considering that the bacterial derived antigens are most probably the relevant antigens.

We thank the reviewers for this valuable comment and have revised the introduction and discussion according to their suggestion.

REVIEWERS' COMMENTS:

Reviewer #1 (Remarks to the Author):

The authors have addressed most of my concerns adequately. Although some of my suggested experiments couldn't be directly performed due to technical constraints, I applaud their effort in providing alternative experiments to address my criticisms. Overall, I am satisfied with the revised manuscript.

Reviewer #2 (Remarks to the Author):

I am fully satisfied with the reply of the authors.

The authors indeed performed new experiments which substantially improve the manuscript. The message that this paper will bring to the scientific community is extremely interesting.

Some very minor points, which could further improve the paper:

- In the abstract, the author should mention the new human data.
- Page 3 line 57: No need to speak about SLE and RA. (See next point).
- Both in the intro and in the discussion, it would be very important to expand the literature regarding IBD and viral infection.
- Page 5 line 83 and line 93. I would rather move '(Armstrong WE)' to line 83.
- Page 6, line 119-121. I believe that there is an extra "and".
- Page 7, line 154. I would remove or change " To our surprise".
- In the discussion, it would be interesting to comment on the fact that non-microbiota specific Treg cells are able to control CD8 microbiota reactive T cells.

Point-by-point reply

Manuscript NCOMMS-19-09838A: “Rapid expansion of Treg cells protects from collateral colitis following a viral trigger”; Schorer et al.

We would like to thank the reviewers and the editors for the constructive review process and are very pleased that our manuscript is in principal suitable for publication in *Nature Communications*. A detailed description on how we addressed the minor points raised by reviewer 2 and the editorial requests is outlined in the point-by-point response below:

REVIEWERS' REQUESTS:

Reviewer #2:

I am fully satisfied with the reply of the authors.

The authors indeed performed new experiments which substantially improve the manuscript. The message that this paper will bring to the scientific community is extremely interesting.

We thank the reviewer for this positive feedback.

Some very minor points, which could further improve the paper:

- *In the abstract, the author should mention the new human data.*

We thank the reviewer for this suggestion and have modified the abstract accordingly.

- *Page 3 line 57: No need to speak about SLE and RA. (See next point).*
- *Both in the intro and in the discussion, it would be very important to expand the literature regarding IBD and viral infection.*

According to the reviewer's suggestion, we have expanded the literature regarding IBD and viral infection in the introduction and discussion. However, we have kept the reference to RA and SLE as well, as they are the prototypic autoimmune diseases associated with a high type I IFN signature.

- *Page 5 line 83 and line 93. I would rather move "(Armstrong WE)" to line 83.*

We thank the reviewer for this comment. We use LCMV strain WE as the standard strain for acute infections throughout the paper and LCMV clone 13 as a standard strain for chronic infections throughout the paper. This is how we initially introduce these two strains (line 87/88). However, in Figure 1D and Supplementary Figure 1D, we also test other LCMV strains that can elicit acute or chronic infections, Armstrong and Docile, respectively. Hence we only mention these strains when describing the experiments depicted in Figure 1D and Supplementary Figure 1D (line 97).

- *Page 6, line 119-121. I believe that there is an extra "and".*

We thank the reviewer for spotting this error and have corrected it in the revised manuscript.

- *Page 7, line 154. I would remove or change "To our surprise".*

We have modified this sentence in the revised manuscript.

- *In the discussion, it would be interesting to comment on the fact that non-microbiota specific Treg cells are able to control CD8 microbiota reactive T cells.*

We thank the reviewer for this suggestion and have included this point in the discussion.